# Single-cell atavism reveals an ancient mechanism of cell type diversification in a sea anemone

Leslie S. Babonis [1,2] ✉, Camille Enjolras [1], Abigail J. Reft [3,4], Brent M. Foster [1], Fredrik Hugosson [1], Joseph F. Ryan [1,5], Marymegan Daly[6] & Mark Q. Martindale [1,5]

Cnidocytes are the explosive stinging cells unique to cnidarians (corals, jelly-fish, etc). Specialized for prey capture and defense, cnidocytes comprise a group of over 30 morphologically and functionally distinct cell types. These unusual cells are iconic examples of biological novelty but the developmental mechanisms driving diversity of the stinging apparatus are poorly character-ized, making it challenging to understand the evolutionary history of stinging cells. Using CRISPR/Cas9-mediated genome editing in the sea anemone *Nematostella vectensis*, we show that a single transcription factor (*NvSox2*) acts as a binary switch between two alternative stinging cell fates. Knockout of *NvSox2* causes a transformation of piercing cells into ensnaring cells, which are common in other species of sea anemone but appear to have been silenced in *N. vectensis*. These results reveal an unusual case of single-cell atavism and expand our understanding of the diversification of cell type identity.

Novel cell types promote the origin of new physiological functions and are essential for the expansion of organismal diversity; yet under-standing the mechanisms that drive the origin of new cell types is a persistent challenge in biology[1,2]. Studies of trait evolution in multi-cellular organisms have identified "homeosis"–the transformation of one part of an organism into another part[3]–as an important source for morphological innovation. Common examples of such homeotic novelties include the conversion of petals into sepals in flowering plants and the transformation of abdominal into thoracic segments in insects[4]. These transformations have since been shown to be con-trolled by regionally expressed regulatory genes[5,6], demonstrating how the development of complex phenotypes can be coordinated by a single genetic switch. More recently, homeosis has been invoked to explain how transformations can occur at the level of a single cell. Studies of neural differentiation in nematodes, flies, and vertebrates have identified individual regulatory genes sufficient to convert one neural subtype into another, supporting the idea that homeosis may

control individuation of cell fate[7]. To date, however, no study has demonstrated homeotic induction of atavism–the reemergence of an ancestral character–at the level of an individual cell type. Such a demonstration would provide a mechanistic understanding of how homeosis could drive cell type diversification and promote the expansion of animal complexity.

Stinging cells (cnidocytes) are among the most morphologically complex of all cell types and are an iconic example of a cellular novelty (Fig. 1). Found exclusively in cnidarians (a clade that includes corals, jellyfish, and their relatives), stinging cells comprise a diverse array of cells specialized for prey capture, defense, and habitat use that vary widely in both form and function across taxa[8]. Indeed, the suite of stinging cell types found in each species of cnidarian is arguably the most informative diagnostic character for soft-bodied taxa, which comprise over 2/3 of the 13,000 species of extant cnidarians[9,10]. Three major subtypes of stinging cells are currently recognized and can be discriminated by the morphology of their stinging organelle[11]. The first,

[1]Whitney Laboratory for Marine Bioscience, University of Florida, St. Augustine, FL, USA. [2]Department of Ecology and Evolutionary Biology, Cornell University, Ithaca, NY, USA. [3]National Systematics Laboratory, Office of Science and Technology, NOAA Fisheries, Washington DC, USA. [4]Department of Invertebrate Zoology, Smithsonian National Museum of Natural History, Washington DC, USA. [5]Department of Biology, University of Florida, Gainesville, FL, USA. [6]Department of Evolution, Ecology, and Organismal Biology, The Ohio State University, Columbus, OH, USA. ✉e-mail: lsb257@cornell.edu

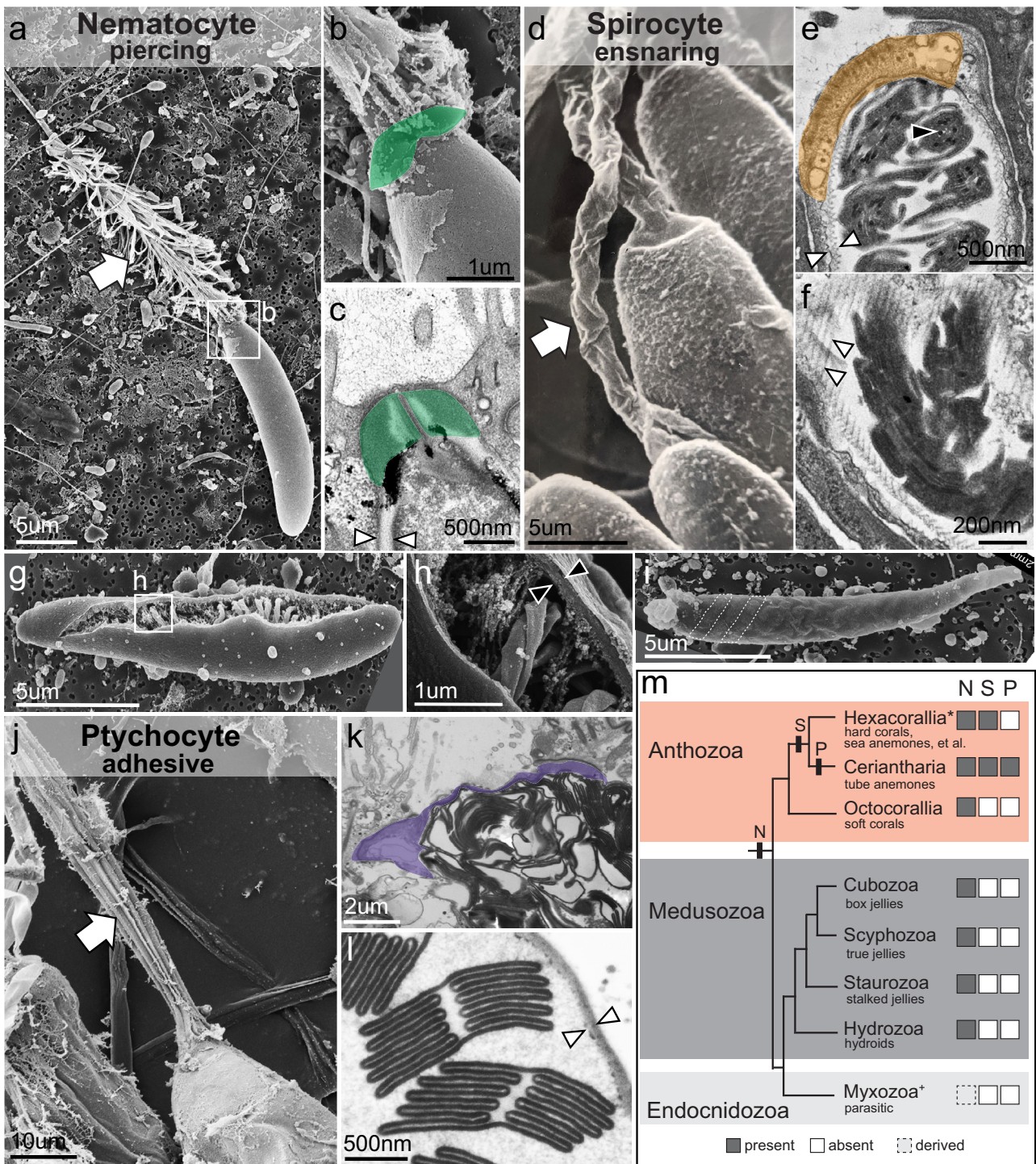

**Fig. 1 | Morphological and phylogenetic diversity of stinging cells.**
**a**, **b** Discharged nematocyte (SEM) from the mesentery of the sea anemone *Nematostella vectensis* showing apical flaps (**b** −green, false colored) and spines along the everted harpoon (white arrow). **c** Apex of undischarged nematocyte (TEM) from the mesentery of *N. vectensis* showing apical flaps (green) and thick capsule wall (arrowheads). **d** Discharged spirocytes (SEM) from the tentacles of the sea anemone *Calliactis tricolor* showing lack of apical flaps and no spines on the everted tubule (white arrow). **e**, **f** Undischarged spirocytes (TEM) from the tentacles of *N. vectensis* showing an apical cap (orange, false colored) and a thin, serrated capsule wall (white arrowheads). The serrated appearance of the spirocyte capsule wall arises from an internal network of regularly spaced fibers (white arrowheads in **f**). Fine lateral rods adorn the tubule and appear as small, dark puncta in cross section (black arrowhead in **e**). **g**, **h** SEMs of a broken nematocyst capsule from *N. vectensis* showing the thick capsule wall (black

arrowheads). **i** Intact spirocyte capsule from *N. vectensis*; the coils of the tubule are visible through the thin capsule wall (two coils are delineated with dashed lines). **j** Discharged ptychocyte (SEM) from the body wall of the tube anemone *Ceriantheopsis americana*; the everted tubule lacks spines but has longitudinal pleats (white arrow). **k** Apex of undischarged ptychocyte from the body wall of *C. americana* showing no specialization (purple, false colored). **l** Cross section of pleated tubule inside the capsule of an undischarged ptychocyte from *C. americana*; the capsule wall is thin and not serrated (white arrowheads). **m** Cladogram of cnidarians, after Kayal et al.[10]; boxes to the right indicate presence (gray) or absence (white) of each type of stinging cell. The hypothesized origins of the three stinging cell types are plotted on the tree. Stippled gray reflects highly derived stinging cells in myxozoans. N nematocyte, S spirocyte, P ptychocyte. *Non-cerianthid hexacorals; this clade does not currently have an accepted name. [+]Myxozoans and *Polypodium*.

nematocytes ("piercing cells") (Fig. 1a–c, g, h), have an explosive organelle with a thick capsule wall containing an eversible, venom-laden tubule[12–15]. Across cnidarians, over 30 different types of nema-tocytes have been described, differing largely in the morphology of the basal, spiny portion of the tubule (hereafter called the "harpoon")[8]. The second stinging cell type, spirocytes ("ensnaring cells"), exhibit much less morphological variation and are described as either gracile (thin) or robust (thick) in overall appearance[16]. While still extrusive, these cells are typified by two traits not found in piercing cells: a thin capsule wall with a serrated appearance, derived from a network of regularly spaced fibers lining the inner capsule wall, and an eversible tubule adorned with fine lateral rods that create an ensnaring web upon discharge[17] (Fig. 1d–f, i). The third group of stinging cells, ptychocytes ("adherent cells"), are not known to contain toxins and their payload consists of only a pleated, sticky tubule folded inside a thin-walled capsule[18] (Fig. 1j–l). The broad distribution of piercing cells (nemato-cytes) throughout the phylum suggests this cell type was present in the last common ancestor of cnidarians (Fig. 1m). The more restricted distribution of ensnaring and adherent cell types (spirocytes and pty-chocytes, respectively) implies that these cell types evolved from a piercing ancestor. However, the developmental mechanisms control-ling variability in the morphology of the stinging apparatus have not been characterized. The evolutionary factors driving the diversification of these unusual cell types, therefore, remain unresolved.

*Sox* genes are a family of transcription factors that play an instructive role in cell fate decisions across animals[19]. Tissue-restricted expression of various *Sox* orthologs has been demonstrated previously in cnidarians and is consistent with a role for these transcription fac-tors in patterning distinct cell types[20–24]. Here, we show that a single *Sox* gene, *NvSox2*, regulates a homeotic switch between two alternative stinging cell fates: piercing and ensnaring. Interpreted in the context of sea anemone phylogeny, these results provide an explanation for the discontinuous distribution of distinct types of stinging cells across cnidarians and, suggest single-cell atavism may be an important mechanism driving the evolution of novelty.

## Results

### Stinging cell distribution in *Nematostella* vectensis

In *N. vectensis*, stinging cells are found throughout the ectodermal epithelia but each region of the animal is populated by a distinct combination of stinging cell types[25]. The tentacle tips, for example, are populated by large and small basitrichous isorhiza nematocytes (piercing cells) and gracile spirocytes (ensnaring cells) whereas the body wall (outer epithelium) is populated exclusively by large and small piercing cells (Fig. 2a, b). The ectodermal epithelia of the mesenteries (digestive tissues) are populated largely by microbasic mastigophores (another type of piercing cell) and the foot is popu-lated almost exclusively by small basitrichous isorhizas (Fig. 2c, d). The distributions and relative abundance of the different stinging cell types are summarized in Fig. 2b.

### Knockout of NvSox2 causes homeotic transformation of sting-ing cells

*NvSox2* is one of 14 *Sox* genes in *N. vectensis*, only two of which (*SoxB2* and *NvSox2*) are expressed in individual cells throughout the ecto-derm, a pattern consistent with specification of terminal cell identity during early embryogenesis[20]. We examined the expression of *NvSox2* further and found it to be co-expressed with the transcription factor *PaxA* in developing stinging cells (Fig. 2e). We identified cells expres-sing *NvSox2* only, cells co-expressing both *NvSox2* and *PaxA*, and cells expressing *PaxA* only, suggesting sequential activation of *NvSox2* and then *PaxA*. Using qPCR in combination with morpholino-mediated gene knockdown, we found significant suppression of *NvSox2* expression following knockdown of *SoxB2* (expressed in progenitors of neurons and stinging cells)[22] and no significant change in *NvSox2*

expression in response to knockdown of *PaxA* (Fig. 2f). Together, these results suggest *NvSox2* is downstream of *SoxB2* and upstream of *PaxA* in early differentiating stinging cells (Fig. 2g). To understand the function of *NvSox2* in stinging cell specification, we used CRISPR/Cas9-mediated genome editing to generate F2 homozygous knockouts for the *NvSox2* locus (Supplementary Fig. 1). *NvSox2* is first expressed at the blastula stage and continues to be expressed throughout early development in individual cells scattered throughout the ectoderm; this expression is completely abolished in *NvSox2* mutants (Fig. 2h, Supplementary Fig. 2). We allowed these *NvSox2* mutants to grow to the polyp stage to investigate the molecular and morphological effects resulting from loss of *NvSox2*.

To understand the molecular role of *NvSox2* in stinging cell fate, we examined the expression of genes known to be specific to stinging cells in wild type and *NvSox2* knockout (mutant) animals[25,26]. Expres-sion of the transcription factor *PaxA* (Fig. 2i, j; Supplementary Fig. 3) and the structural molecule minicollagen-1 (*Mcol1*; Fig. 2k, l) (both piercing cell-specific) were significantly reduced in the body wall of *NvSox2* mutants. By contrast, minicollagen-4 (Mcol4), which is found in all types of stinging cells, was unaffected. Quantitative analysis of stinging cell development revealed that 80% of the Mcol4-labeled cells in the body wall of wild type polyps also express *Mcol1* and this pro-portion is significantly reduced to <10% in *NvSox2* mutants (Fig. 2l). These results confirm that knockout of *NvSox2* did not affect the total number of stinging cells specified in the body wall but transformed the identity of the stinging cells in this tissue.

In *N. vectensis*, the body wall of wild type polyps is populated almost exclusively by a single type of piercing cell, the small basi-trichous isorhiza (Fig. 3a)[27,28]. To investigate the effects of *NvSox2* knockout on the development of body wall stinging cells further, we examined the morphology of these cells using light and electron microscopy. In *NvSox2* mutants the small piercing cells of the body wall were completely replaced by a morphologically distinct cell type (Fig. 3b). At a gross level, the mutant cell type had the appearance of a robust spirocyte (ensnaring cell), a type of stinging cell that is not normally found in *N. vectensis* but is common in other sea anemones (**Source Data**). We investigated these mutant cells for evidence of ensnaring cell features using scanning and transmission electron microscopy (SEM and TEM, respectively). Stinging cells from the body wall of wild type polyps had rigid capsules with a thick capsule wall and prominent flaps at the apical (outward facing) end (Fig. 3c–e). When discharged, these stinging cells revealed an extrusive tubule with large spines proximally (the "harpoon") and small barbs distally (Fig. 3f). These features are diagnostic of sea anemone piercing cells[25,29,30]. By contrast, the capsule of the extruded mutant cells appeared flimsy, collapsing upon discharge, and the extrusive apparatus consisted of a smooth tubule devoid of spines and barbs (Fig. 3g). The mutant cells also have a thin capsule wall with a serrated appearance along the inner surface and a flat apical cap (Fig. 3h–j), features consistent with ensnaring cells, not piercing cells[17,31,32]. Furthermore, cross-sections of the internalized extrusive tubule in the mutant cells revealed small rods consistent with the appearance of undischarged ensnaring cells (Fig. 3k), although these rods were smaller and fewer in number than has been described for the ensnaring cells found in wild type animals[17]. Without cell lineage labeling, we cannot rule out the possibility that knockout of *NvSox2* caused both the loss of piercing cells and the gain of ensnaring cells in two distinct cell lineages in the body wall of *N. vectensis*. Considering loss of *NvSox2* did not affect the number of stinging cells expressing Mcol4 in the body wall, the simplest expla-nation for these results is that loss of *NvSox2* causes a transformation of small piercing cells into ensnaring cells in the body wall of *N. vectensis*.

### Knockout of NvSox2 restores an ancestral cell type

Despite having the canonical morphological features of mature ensnaring cells, the cells that developed in the body wall of *NvSox2*

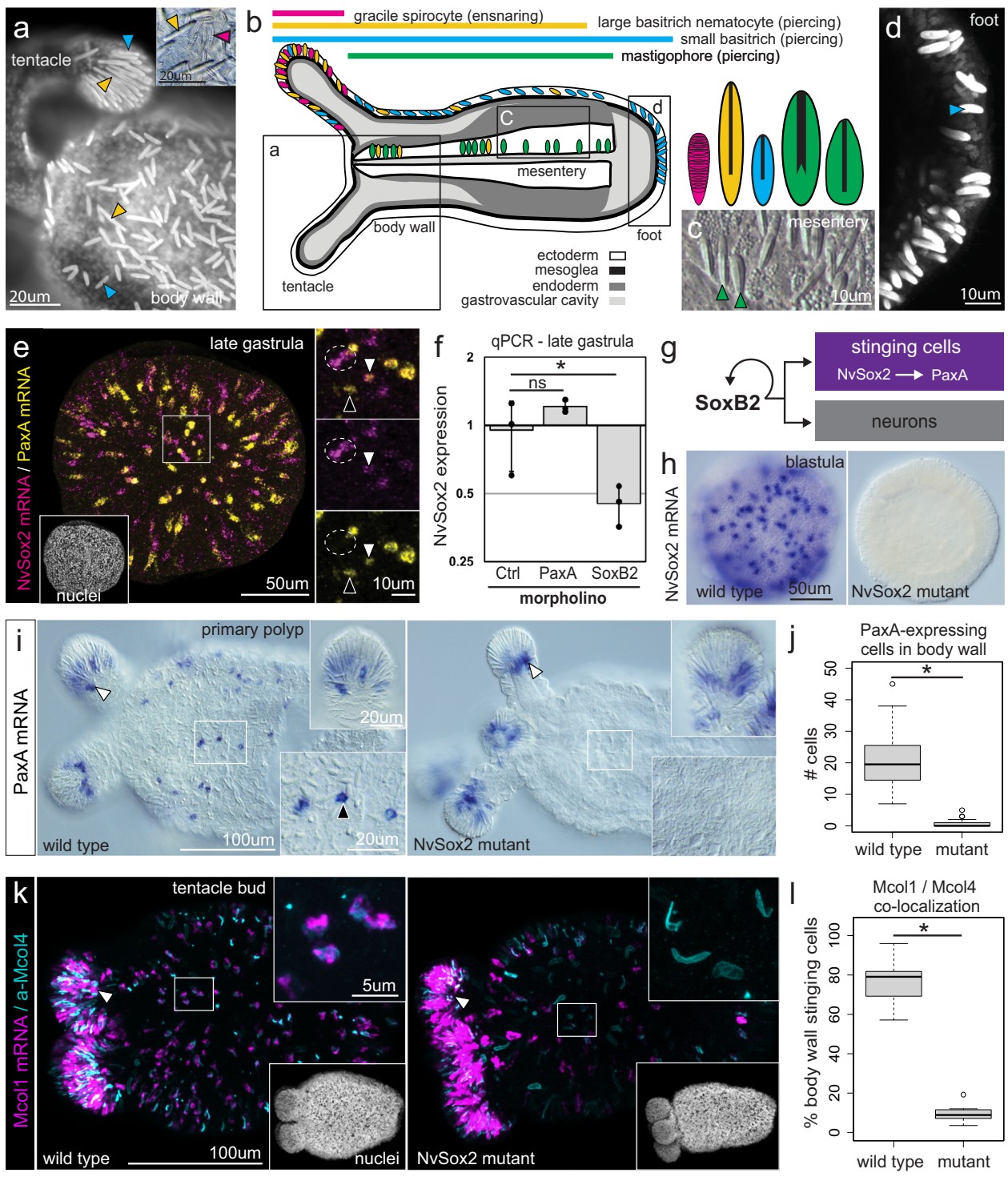

mutants were morphologically distinct from the gracile ensnaring cells that develop in the tentacle tips of wild type animals, leading us to hypothesize that loss of *NvSox2* caused the development of robust ensnaring cells in the body wall of *N. vectensis*. The "robust" and "gracile" conditions are not well-defined; to remedy this, we analyzed the length and width of the capsule from undischarged ensnaring cells from the literature reported to be either robust or gracile. The relationship between capsule width and length did not differ for robust and gracile ensnaring cells across a range of sizes (Fig. 3l); however, for a given length, robust ensnaring cells were significantly wider than gracile cells (Fig. 3l, m). The width of the capsule that developed in the

body wall of *NvSox2* mutants is distinct from and intermediate between the capsule width reported for gracile and robust ensnaring cells in the literature (Fig. 3m). Surprisingly, examination of wild type animals revealed the presence of cells with this mutant morphology, albeit in very low frequency. While the mutant cell type comprised nearly 50% of the stinging cells in *NvSox2* mutants, fewer than 1% of the stinging cells of wild types had this morphology (Fig. 3n). Of the 12 wild type polyps examined, only one appeared to lack these mutant cells completely. These results suggest errors may arise spontaneously in the *NvSox2* regulatory pathway at low frequency resulting in the development of robust ensnaring cells in wild type animals. Because

**Fig. 2 | NvSox2 is a stinging cell regulatory gene. a** Primary polyp labeled with 143uM DAPI showing nuclei and mature nematocyte capsules (*N* = 10 polyps); small (blue) and large (yellow) basitrichs are indicated. Inset: gracile spirocytes (magenta) and large basitrichs (yellow) are abundant in the tentacles. **b** Distribution of stinging cell types in *N. vectensis*; illustration modified from: Babonis, L. S., Ryan, J. F., Enjolras, C. & Martindale, M. Q. Genomic analysis of the tryptome reveals molecular mechanisms of gland cell evolution. *EvoDevo* **10**, 23 (2019)−CC BY 4.0. **c** Mesenteries are populated largely by mastigophores (green) (*N* = 12 polyps). **d** Small basitrichs dominate the body wall and foot (143uM DAPI) (*N* = 10 polyps). **e** *NvSox2* and *PaxA* overlap in expression (*N* = 8 embryos); white arrowheads: cells expressing both, black arrowheads: *PaxA* only, dotted circles: *NvSox2* only. **f** *NvSox2* is significantly downregulated in response to *SoxB2* knockdown (ddCT method, *p* = 1.12E-2) but is not significantly affected by knockdown of *PaxA* (ddCT, *p* = 0.345957) (qPCR). Error bars indicate + /− standard deviation. Three replicate experiments for each treatment are shown (black dots); gray bars indicate averages of the three experiments. **g** Working model. **h** *NvSox2* expression is abolished in *NvSox2* mutants. **i** *PaxA* expression in the tentacles (white arrowhead and inset) and body wall (black arrowhead and inset) of wild types and mutants (white box and inset). **j** Quantitative analysis of cells expressing *PaxA* in the body wall (Mann–Whitney U, *p* = 1.37E-11). *N* = 32 (wild type) and *N* = 30 (mutant) animals were examined per treatment (primary polyp stage). **k** *Mcol1* mRNA (magenta) is co-expressed with Mcol4 protein (α-Mcol4, cyan) in the tentacles (white arrowhead) and body wall (white box and inset) of wild types; *Mcol1* expression is reduced in the body wall of *NvSox2* mutants. Nuclei−white, DAPI. **l** Quantitative analysis of cells in the body wall co-expressing *Mcol1* and Mcol4 (Mann–Whitney U, *p* = 1.57E-4). *N* = 10 animals per treatment (tentacle bud stage). Oral pole to the left in **a**, **e**, **i**, **k**. For all box plots: median−middle line, 25th and 75th percentiles−box, 5th and 95th percentiles−whiskers, outliers−individual points. Source data are provided as a Source Data file. Fluorescent images were adjusted for contrast and brightness using Fiji.

robust ensnaring cells are common and abundant in other sea anemones (Source Data), it is reasonable to suggest that this cell type was likely present in the most recent common ancestor of all sea anemones. Therefore, our results suggest knockout of *NvSox2* has resulted in single-cell atavism, or the restoration of an ancestral cell type (robust ensnaring cell) in *N. vectensis*.

To further investigate the identity of ensnaring cells, we examined the distribution and development of the native gracile ensnaring cells and the mutant ensnaring cells in *N. vectensis*. In both wild type and *NvSox2* mutant animals, gracile spirocytes were found only in the tentacle tips (Fig. 4a); by contrast, small piercing cells were present in the tentacle tips of wild type animals only and mutant ensnaring cells were present in the tentacles of *NvSox2* mutant animals only. These results are consistent with the transformation of small piercing cells into mutant cells seen in the body wall (Fig. 3), where small piercing cells are the predominant cell type. To determine if gracile and mutant ensnaring cells are molecularly distinct, we examined the expression of a Ca2+-binding protein called Calumenin F (*CaluF*)[33]. In wild type animals, *CaluF* is expressed exclusively in the tentacles, consistent with the distribution of gracile cells, and is co-expressed in cells labeled with anti-Mcol4 antibody[25], confirming it is expressed in stinging cells (Fig. 4b). However, *CaluF* is never co-expressed with the piercing cell-specific gene, *Mcol1* (Fig. 4B)[25] and is therefore a positive and specific marker of gracile ensnaring cells. Knockout of *NvSox2* did not affect the timing of appearance or the distribution of cells expressing *CaluF* (Fig. 4c), confirming the mutant ensnaring cells are not gracile cells. Furthermore, analysis of gracile cell morphology by TEM confirmed that gracile cells from wild type and mutant animals are morphologically indistinguishable (Fig. 4d−i). In light of these results, we suggest that knockout of *NvSox2* causes a homeotic transformation of small piercing cells into robust ensnaring cells anywhere small piercing cells would normally be specified. By contrast, loss of *NvSox2* did not affect the development of gracile ensnaring cells, suggesting gracile and robust ensnaring cells are patterned using distinct mechanisms.

## NvSox2 also controls harpoon morphogenesis

In *N. vectensis*, the wild type tentacle tip epithelium is populated by small piercing cells (rare), large piercing cells (abundant), and gracile ensnaring cells (abundant) (Fig. 2b)[25,26,28]. Examination of the tentacle tips by light microscopy confirmed that the small piercing cells were completely transformed into robust ensnaring cells, mirroring the effects in the body wall, but large piercing cells are present in both wild type and mutant animals (Fig. 5a). Large piercing cells initially appeared unperturbed by the loss of *NvSox2* as there was no detectable change in the distribution of these cells or the expression of *PaxA* or *Mcol1* in the tentacle tips (Fig. 2i, k); however, using an H2O2-conjugated tyramide to label the harpoons of piercing cells, we show the mutant animals exhibit aberrant harpoon morphology (Fig. 5b). Specifically, the proximal portion of the harpoon appeared looped in the large piercing cells of mutant animals, as compared to the straight harpoon found in wild type polyps. To assess changes in the abundance of piercing cells in the tentacle tip, we counted the number of labeled harpoons relative to the total number of nuclei and show a significant loss (~20%) of piercing cells in mutant animals (Fig. 5c). This result is consistent with the transformation of the small piercing cells in the tentacle tips into robust ensnaring cells (Fig. 5a) and suggests small piercing cells comprise ~20% of the piercing cells in the tentacle tip of wild type animals. To further investigate their aberrant harpoon morphology, we induced discharge of tentacle tip piercing cells in both wild type and mutant polyps using an infrared laser ablation system (XY Clone, Hamilton Thorne, USA) (Supplementary Movies 1, 2). Examination of the discharged piercing cells suggested that the looped phenotype of the mutants is due to elongation of the harpoon (Fig. 5d, e). To quantify this effect, we measured the length of the harpoon and the length of the capsule in discharged stinging cells from the tentacle tips of wild type and mutant polyps and show a significant increase (~50%) in the length of the harpoon in mutant animals (Fig. 5f).

The looped harpoon of the large piercing cells in the tentacle tips of *NvSox2* mutants phenocopies the looped appearance of the harpoon in a type of piercing cell found in other species of sea anemone: the macrobasic mastigophore[34]. In macrobasic piercing cells, the harpoon is longer than the capsule in which it is contained so the harpoon must loop to fit inside the capsule; this condition is demonstrated by a macrobasic mastigophore from the sea anemone *Cylista elegans* (Fig. 5g). By contrast, microbasic cells develop a harpoon that is shorter than the length of the capsule so the harpoon fits completely inside the capsule without looping. In *N. vectensis*, all the mastigophores are microbasic and are found only in the ectodermal epithelia of the digestive tract (Fig. 2b). Knockout of *NvSox2* did not affect the size of the mastigophore harpoons in *N. vectensis* as all mastigophores retained the microbasic morphology in both wild type and mutant animals (Fig. 5h). While the role of *NvSox2* in driving the elongation of the harpoon appears to be restricted to the large piercing cells (basitrichous isorhizas) in *N. vectensis*, this single-gene control of harpoon development (Fig. 5I) provides a mechanism by which the macrobasic harpoon condition could have evolved numerous times, independently, in other types of stinging cells[35].

## NvSox2 is restricted to taxa with ensnaring cells

*Sox* genes are ubiquitous among animal genomes and cluster into 8 major groups (*SoxA-SoxH*)[19], at least four of which were likely present in the common ancestor of cnidarians and vertebrates (*SoxB, SoxC, SoxE, SoxF*). Previous phylogenetic analyses of *Sox* genes from *N. vectensis* have suggested that *NvSox2* is either a cnidarian-specific ortholog[20] or a member of the *SoxB* clade[24,36,37]. To test these hypotheses further, we built a maximum likelihood phylogeny of *Sox* genes from 15 cnidarians and 6 bilaterians (Supplementary Fig. 5; Supplementary Data 1). With this extensive taxon sampling, we find strong

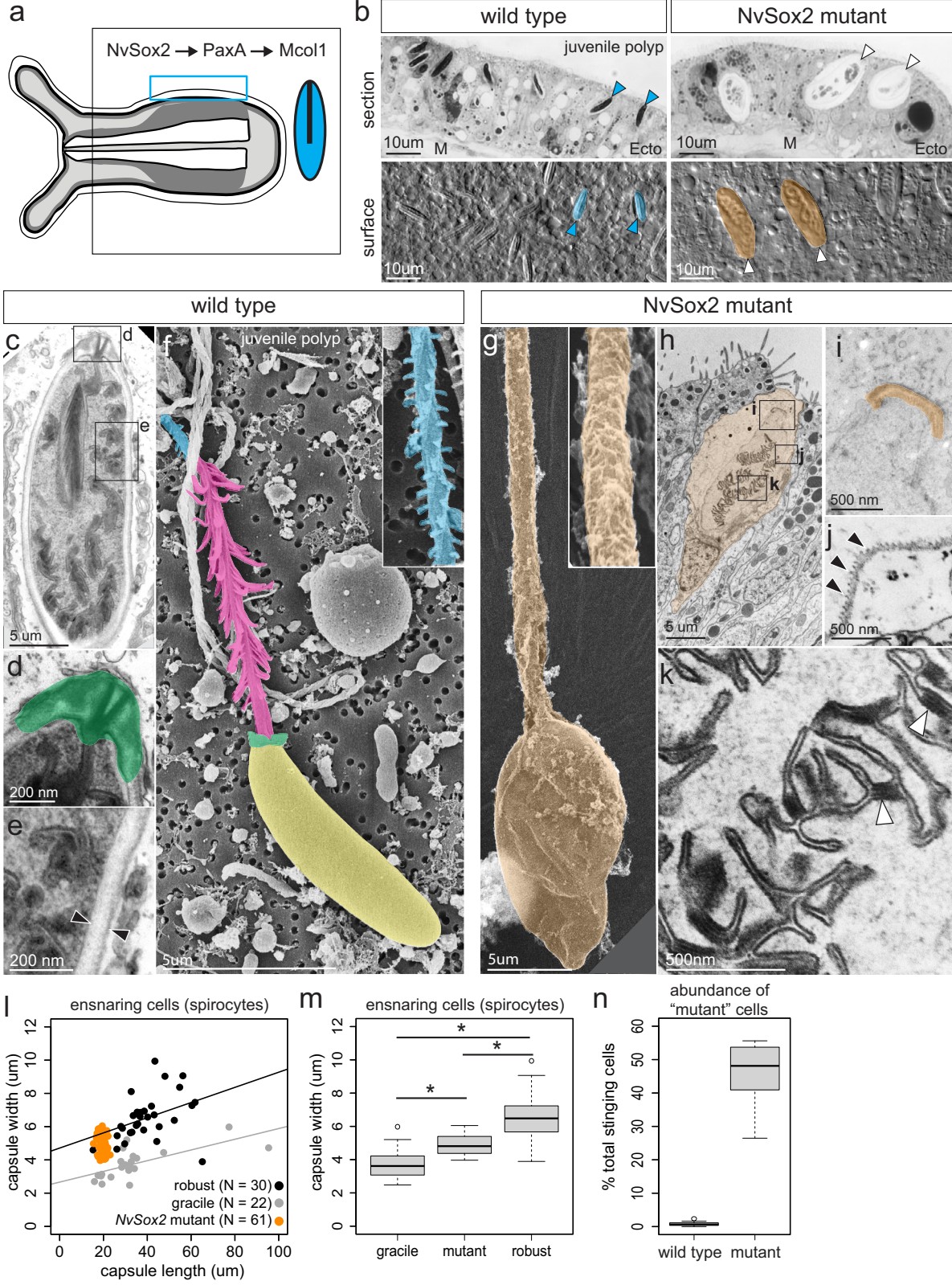

support for the monophyly of a clade including *NvSox2* orthologs from corals and sea anemones but weak support linking the *NvSox2* clade to any known bilaterian *Sox* genes. Additionally, we find no evidence of an ortholog of *NvSox2* in medusozoans, octocorals, or cerianthids (tube anemones), suggesting *NvSox2* arose in the stem ancestor of corals and sea anemones (non-cerianthid hexacorals), after the origin of ensnaring cells (spirocytes) (Fig. 1m).

## Single-cell homeosis explains stinging cell diversity

Although small and large piercing cells have been assumed to be two size classes of the same cell type[25], we show that these are actually two molecularly distinct types of stinging cells. In the absence of *NvSox2* the small piercing cells took on a new (ensnaring) identity (Fig. 6a), but loss of *NvSox2* was insufficient to change cell identity in large piercing cells. Thus, *NvSox2* is required for the development of a suite of traits

**Fig. 3 | Knockout of *NvSox2* causes a homeotic shift in stinging cell identity.**
**a** *NvSox2, PaxA,* and *Mcol1* are required for development of small piercing cells (blue) in the body wall; illustration modified from: Babonis, L. S., Ryan, J. F., Enjolras, C. & Martindale, M. Q. Genomic analysis of the tryptome reveals molecular mechanisms of gland cell evolution. *EvoDevo* **10**, 23 (2019)−CC BY 4.0. **b** Blue arrowheads: small piercing cells; white arrowheads: mutant cells. M: mesoglea, Ecto: ectoderm. Cells are false colored. Images correspond to the area inside the blue box in a; *N* = 2 juvenile polyps per treatment. **c−k** Electron micrographs of stinging cells from the body wall of wild type **c−f** and mutant **g−k** juvenile polyps (*N* = 2 individuals per treatment). **c−e** TEMs of a small piercing cell. Boxes in c indicate regions highlighted in d and e. Apical flaps are false colored in d. Black arrowheads in e highlight the thick capsule wall. **f** SEM of a small piercing cell. Relevant structures are false colored: yellow−capsule, green−apical flaps, magenta −harpoon with spines, blue−tubule with barbs. Inset in f shows detail of tubule. **g** SEM of a mutant stinging cell lacking spines and barbs. **h−k** TEMs of mutant stinging cells. **h** Boxes indicate regions highlighted in i−k. High magnification images show **i** apical cap (false colored orange), **j** thin, serrated capsule wall (back arrowheads), and **k** internalized tubule with short lateral rods (white arrowheads). **l** Capsule morphometrics (width vs length) in gracile (gray) and robust (black) ensnaring cells (extracted from the literature) and mutant cells (orange) (ANCOVA, gracile vs robust: $p = 0.42032$, gracile vs mutant: $p = 0.30121$). $N$ = number of cells analyzed. **m** Capsule width in gracile, mutant, and robust ensnaring cells (ANOVA with Bonferroni post hoc, gracile vs mutant $p = 5.2E-06$; robust vs. mutant $p = 1.3E-11$; gracile vs robust $p = 2E-16$). Averages calculated from the same data presented in panel l. **n** Abundance of cells with the mutant phenotype in wild type and mutant polyps (Mann−Whitney U, $p = 3.2E-5$); $N$ = 12 primary polyps per treatment. For all box plots: median−middle line, 25th and 75th percentiles−box, 5th and 95th percentiles−whiskers, outliers−individual points. Source data are provided as a Source Data file.

associated with piercing phenotype in small piercing cells (e.g., basal spines and a thick capsule with apical flaps) (Fig. 3) and only a single trait (harpoon length) in large piercing cells (Fig. 5). Evolutionarily, the integration of *NvSox2* into the robust ensnaring cell lineage seems to have permitted the parallel origin of piercing cell phenotype (Fig. 6b). Critically, this suggests that piercing cells ("nematocytes") and ensnaring cells ("spirocytes") are not as phylogenetically distinct as once thought, but instead represent two alternative stinging cell fates controlled by a single gene.

Homeotic transformation of cell identity explains how the complement of stinging cells found in any two closely related cnidarians could differ quite dramatically. Burrowing sea anemones are found in a wide variety of habitats and provide a valuable illustration of this phenomenon (Fig. 6c). While some species are found in a sandy habitat (like *Nematostella vectensis* and *Edwardsiella ignota*), others burrow into sea ice (like *Edwardsiella andrillae*)[38]. An evolutionary transition from the robust ensnaring cells found in the body wall of *E. andrillae* to the small piercing cells found in the body wall of *E. ignota* could be explained by incorporation of the *NvSox2* ortholog into the gene regulatory network driving the development of body wall stinging cells in *E. ignota* but not *E. andrillae*. Likewise, the sister species *E. lineata* and *E. carnea* both undergo an ecological transition during their lifetime from a free-living state (burrowing in sand/mud) to a parasitic state (burrowing in the digestive tract of a ctenophore);[39] this transition is accompanied by a transformation of stinging cell types in the body wall at these two different life stages. Piercing cells are found in the body wall of free-living individuals; while the specific identity of the stinging cells in the body wall of these animals during their parasitic stage has been debated[40,41], they are very similar in morphology to the robust ensnaring cells identified in our *NvSox2* mutants. This suggests a similar single-cell homeotic event may regulate the shift in stinging cell identity associated with the variable habitats used by each life stage. This scenario could also explain the numerous transitions among distinct stinging cell types found in the body wall of other burrowing sea anemones that invade unique habitats, providing an explanation for the discontinuous variation in stinging cell types found among closely related species throughout Cnidaria (Fig. 6d)[40,41].

## Discussion

We have shown that knockout of a single transcription factor (*NvSox2*) caused piercing cells to take on the morphology (Fig. 3) and molecular signature (Figs. 2 and 4) of ensnaring cells and that these mutant cells retain the ability to discharge. Together, these results confirm that this mutation is functionally relevant and is, therefore, a bona fide homeotic transformation[7]. Importantly, the transformation resulted in restoration of a type of ensnaring cell that was lost in the ancestor of the lineage that gave rise to *Nematostella vectensis*. Thus, we have shown a special case of homeosis: single-cell atavism, or the restoration of an ancestral cell type in a modern species. As this is the first

evidence of atavism in a cnidarian, our results suggest that homeotic control of cell fate is an ancient mechanism that has contributed to the expansion of biodiversity since the last common ancestor of cnidarians and bilaterians, over 800MYA[42].

Our ability to resurrect an ancestral cell type by manipulating a single gene demonstrates how homeosis links developmental flexibility with environmental selection pressures to drive the emergence of novel cell types. Specifically, the ability to silence the functional properties of ancestral cell types while retaining the instructions for their specification allows animals extreme flexibility while exploring new habitats. As further support of this idea, a single regulatory gene was found to control a homeotic shift in butterfly wing color pattern, providing a mechanism by which rapid morphological evolution can occur when a multicellular phenotype is controlled by a single genetic switch[43]. These examples demonstrate how the ability to select from among multiple possible cell identities with a single gene can create efficient opportunities for adaptation to local conditions. The ability to redeploy silenced cell types may, therefore, be a widespread, but underappreciated, macroevolutionary mechanism for generating cell type diversity in animals.

The observation that *NvSox2* controls harpoon morphogenesis− not specification−in large piercing cells recapitulates phenotypic variation observed in another lineage of stinging cells (mastigophores; Fig. 5). The fact that *NvSox2* controls only the length of the harpoon in large piercing cells suggests it should be possible to find individual genes that similarly control the morphology of the spines, the number of barbs, the composition of the capsule wall, and the shape of the apical structures. Indeed, a recent study of piercing cell function in *N. vectensis* identified a spinalin-like protein[44] as a specific component of the harpoon[45]. Future studies investigating the regulatory relationships of transcription factors, like *NvSox2*, and effector genes driving harpoon morphology, including spinalin-like, on a single-cell basis will provide a unique opportunity to reconstruct the evolutionary diversification of each individual stinging cell type and lead to a deeper understanding of how gene regulatory networks evolve. Similar to the modular control of neural identity described from nematodes[46], this framework explains how evolution could mix-and-match genes controlling fine aspects of subcellular phenotype to produce the diverse array of cell types that drive animal biodiversity and provides a model for characterizing the evolution of other novel organelles.

## Methods
### Genome editing
The *NvSox2* locus was deleted using a modification of two published CRISPR/Cas9 protocols[47,48]. Briefly, five guide RNAs were designed to target different sites in the *NvSox2* locus, four in the coding sequence and one in the 5' UTR (Supplementary Fig. 1, Table 1). Guide RNAs were purchased from Synthego (USA) and reconstituted to 30 uM in nuclease-free water (Synthego) following the manufacturer's

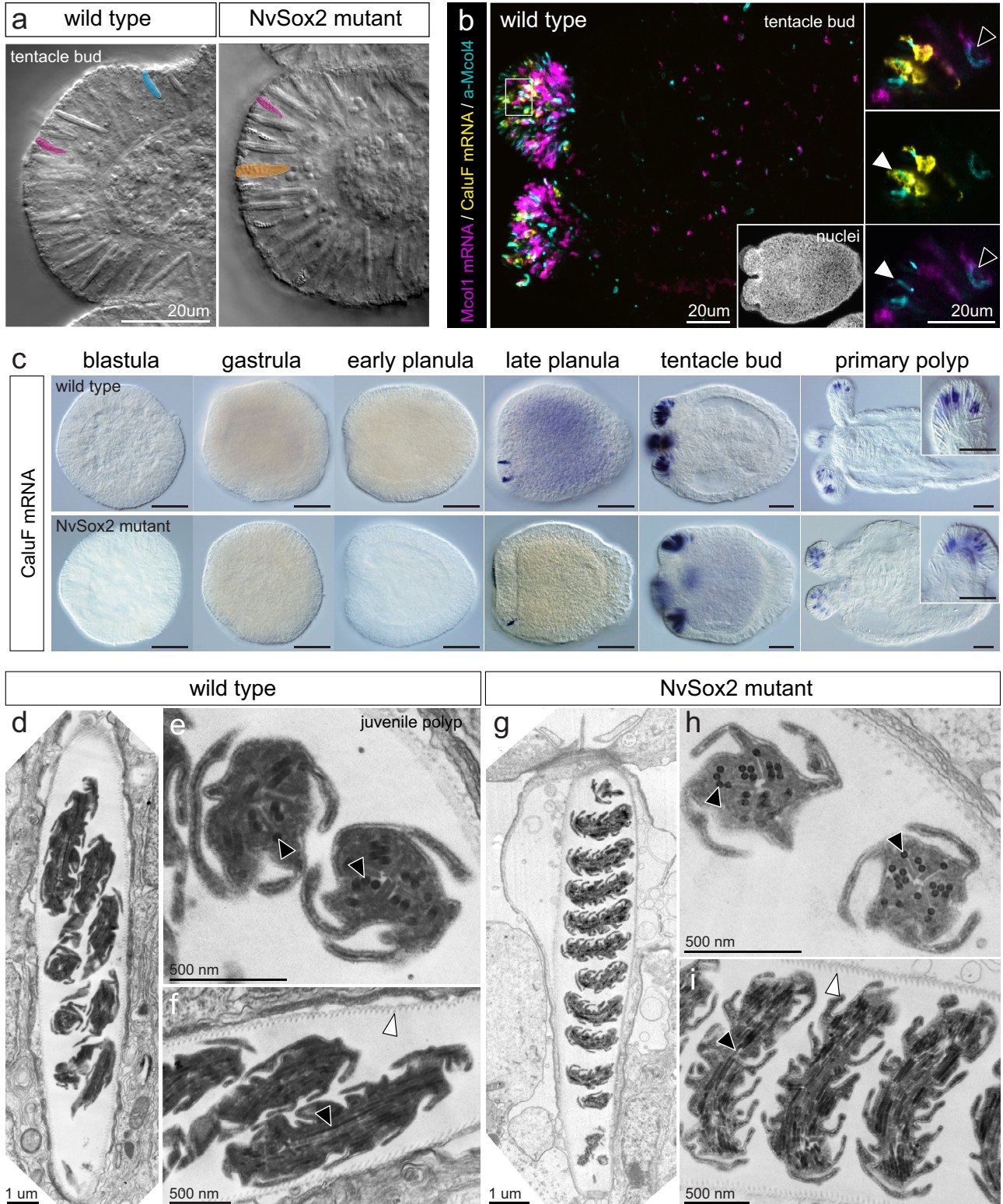

**Fig. 4 | Loss of *NvSox2* does not affect the development of gracile ensnaring cells (spirocytes). a** Gracile ensnaring cells (false colored magenta) are present in the tentacle tips of both wild type (*N* = 5) and NvSox2 mutant (*N* = 4) animals; small piercing cells (blue) are only in wild types and robust ensnaring cells (orange) are only in mutants. Colors correspond to Fig. 2b. Oral pole to the left in **a**–**c**. **b** *CalumeninF* (*CaluF*) is a specific marker of gracile ensnaring cells. Mcol4 (cyan) and *Mcol1* (magenta) are co-expressed in the tentacles and body wall (black arrowhead; inset) (*N* = 6 animals). *CaluF* (yellow) is expressed only in the tentacles.

Insets: *CaluF* is co-expressed with Mcol4 (white arrowheads) but never with *Mcol1* (magenta). **c** Knockout of *NvSox2* did not affect the onset of *CaluF* expression or the distribution of *CaluF*-expressing cells. All scale bars in **c**: 50 um. Each panel is representative of at least 10 individuals at each stage. **d**–**i** TEMs of gracile ensnaring cells from wild type (*N* = 3) and mutant (*N* = 2) juvenile polyps. Knockout of *NvSox2* did not affect the morphology of the serrated capsule wall (white arrowheads) or the lateral rods on the internalized tubule (black arrowheads). Fluorescent images were adjusted for contrast and brightness using Fiji.

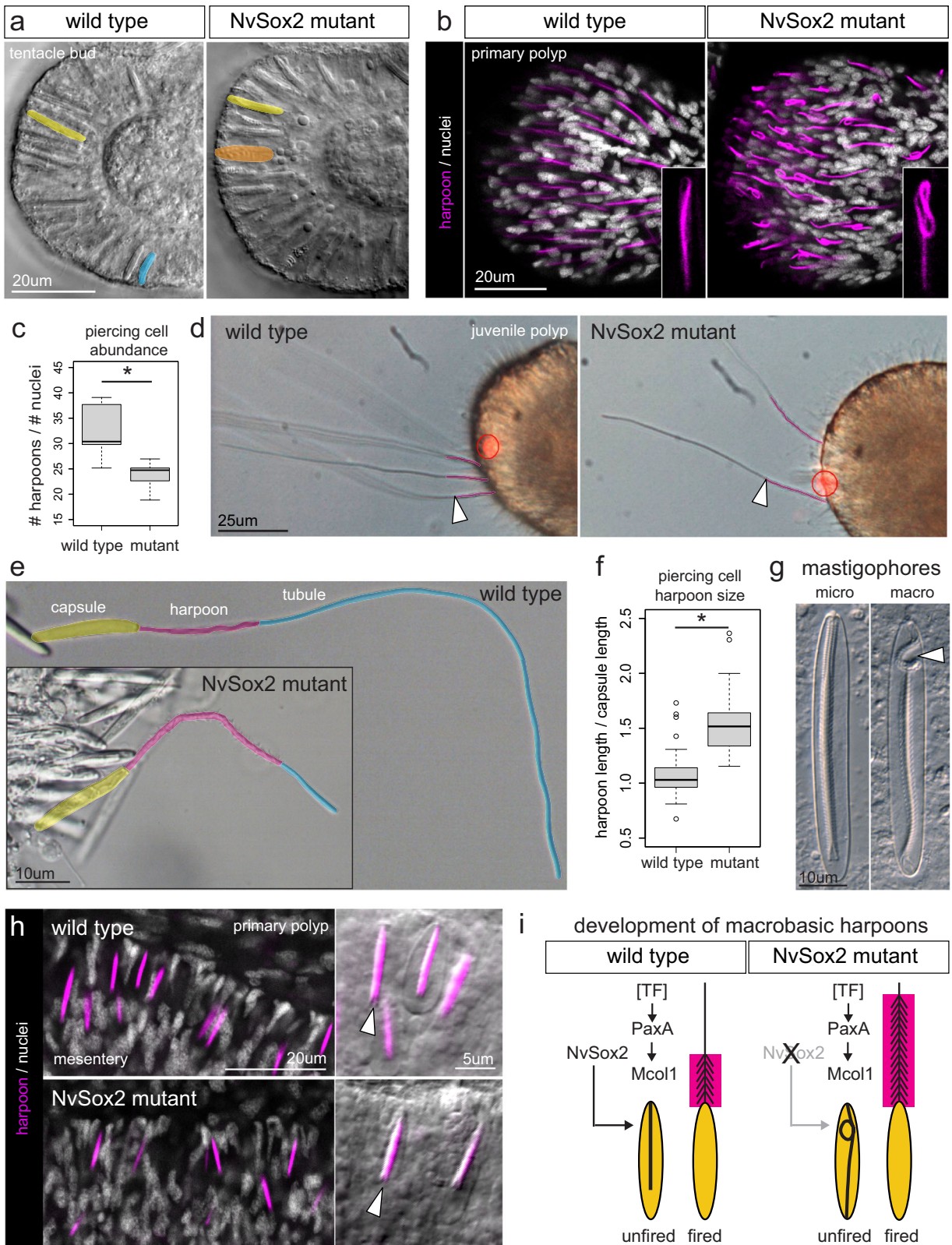

instructions. Cas9 protein (PNABio CP01-50) was reconstituted to 1 mg/ml in nuclease-free water (Ambion AM9937) following the manufacturer's instructions. Guide RNAs were mixed together with Cas9 in a ratio of 5:4 (vol:vol) gRNA:Cas9, incubated at room temperature for 10 min, and injected into zygotes with 0.2 mg/ml Alexa-555 RNAse free dextran (Invitrogen D34679) and nuclease-free water (Ambion). To determine if there were non-specific effects of guide RNAs or Cas9,

control embryos were injected with guide RNAs but no Cas9 protein or with Cas9 protein alone. These animals developed normally and were not examined further. Mosaic F0 *NvSox2* mutant embryos were raised to reproductive maturity at 16 C and spawned to generate the F1 generation. Genomic DNA was extracted from tentacle tips of F1 polyps and sequenced to check for mutations. One F1 female and one F1 male were identified to have significant deletions in the *NvSox2*

**Fig. 5 | *NvSox2* affects harpoon morphogenesis in large piercing cells. a** Large piercing cells (false colored yellow) are present in the tentacle tips of both wild type (*N* = 5) and *NvSox2* mutant (*N* = 4) animals; small piercing cells (blue) and robust spirocytes (orange) are also indicated. Colors correspond to Fig. 2b. **b** Harpoons (magenta) from large piercing cells in *NvSox2* mutants have aberrant (looped) morphology. Nuclei−white (DAPI). **c** Piercing cell abundance is significantly lower in the tentacle tips of *NvSox2* mutants than in wild types (Mann−Whitney U, *p* = 1.6E-4). *N* = 11 primary polyps per treatment. **d** Still images from videos of laser-induced stinging cell discharge (Supplementary Movies S1 and S2). White arrowheads denote transition from harpoon to tubule. Red circled area shows the laser target (moved after discharge). **e** Large piercing cells after induced discharge; capsule, harpoon, and tubule are false colored. Tubules did not fully discharge and appear short in mutants. **f** Harpoons are significantly longer in large piercing cells from mutant animals than wild types

(Mann−Whitney U, *p* = 2.1E-13). Number of cells analyzed: *N* = 57 (wild type), *N* = 41 (mutant). **g** Looped harpoon morphology (white arrowhead) is present in the macrobasic mastigophores from the sea anemone *Cylista elegans* (*N* = 2 individuals); a microbasic mastigophore from the same species is also shown. **h** The morphology of the microbasic mastigophores (piercing cells) in the mesenteries of *N. vectensis* was not affected by knockout of *NvSox2* (*N* = 6 animals each). Harpoons (white arrowheads) are labeled as in **b**. **i** Diagram of the role of *NvSox2* in driving harpoon morphogenesis in large piercing cells. Knockout of *NvSox2* does not affect expression of *PaxA* or *Mcol1* in this cell lineage but does increase the length of the harpoon (magenta box). Elongated harpoons must be looped to fit inside the capsule. For all box plots: median−middle line, 25th and 75th percentiles−box, 5th and 95th percentiles−whiskers, outliers−individual points. Source data are provided as a Source Data file. Fluorescent images were adjusted for contrast and brightness using Fiji.

---

locus (Supplementary Fig. 1) and were used to produce the F2 generation. All further cell/tissue analyses were performed in the F2 generation.

### Immunofluorescence and in situ hybridization

To assay the effects of *NvSox2* knockout on stinging cell development, immature stinging cells were labeled overnight at 4 C with an antibody directed against minicollagen-4 (α-Mcol4)[25,26] diluted 1/1000 in normal goat serum (Sigma G9023) (Supplementary Fig. 4) and mature stinging cells (piercing cells only) were labeled for 30 mins at 25 C in high concentration DAPI (143uM in PBS with 0.1% Tween and 10 mM EDTA)[25,26,49]. Embryonic expression of *NvSox2*, *PaxA*, *Mcol1*, and *CaluF* was examined using in situ hybridization (ISH) following established protocols[50,51]. To achieve two-color fluorescent ISH, samples were simultaneously incubated in both digoxygenin- and fluorescein-labeled mRNA probes overnight at 63 C. Probes were then detected sequentially using anti-DIG/POD and anti-FL/POD antibodies (Roche 11207733910, 11426346910) and fluorescein- or Cy3-coupled tyramide.

### Analysis of Mcol1 and Mcol4 co-expression

Developing animals were fixed for examination at the tentacle bud stage (240 h post fertilization at 16 C). To co-localize *Mcol1* mRNA and Mcol4 protein, we performed fluorescent ISH followed by immunohistochemistry in the same tissues[26]. Labeled tissues were then mounted in glycerol on glass slides, imaged with a Zeiss 710 confocal microscope, and z-stacks were rendered into 3D images using Imaris software v 7.6.1 (Oxford Instruments, USA). Regions of interest were demarcated using the crop tool in Imaris and labeled cells were counted by eye in a 100 × 100 um square region of interest aboral to the tentacle buds. We compared the number of cells labeled with α-Mcol4 in wild type and *NvSox2* mutants using a 2-sided Mann−Whitney *U*-test and found no significant difference in the total number of α-Mcol4-labeled cells (*p* = 0.623176). We then counted the number of cells in the same region of interest that co-expressed *Mcol1* and Mcol4 and found a significant decrease in *NvSox2* mutants, relative to wild type animals (*p* = 1.57E-4). Data are presented (Fig. 2l) as percent of cells labeled with α-Mcol4 antibody that were also labeled with *Mcol1* mRNA probe. *N* = 10 tentacle bud stage animals were examined in each treatment.

### Functional genomics and qPCR

To determine if *NvSox2* was part of the stinging cell gene regulatory network, we knocked down *SoxB2* and *PaxA* using morpholinos (GeneTools, LLC) that have been validated previously[22,26] (sequences provided in Table 1). Morpholinos were reconstituted to 1 mM in nuclease-free water (Ambion) and stored in the dark until the day of use, following the manufacturer's instructions. On the day of injection, morpholinos (MOs) were heated to 65 C for 10 mins, centrifuged at 25 C for 1 min, and diluted to 0.9 mM in nuclease-free water with fluorescent dextran to a final concentration of 0.2 mg/ml. For qPCR

analysis, three replicates of each condition (wild type/uninjected, control MO, SoxB2 MO, and PaxA MO) representing three independent injections performed on different days were compared using the delta-delta CT method and the PCR package in the R statistical computing environment[52,53]. Expression values for qPCR analysis are presented as fold-change, relative to the expression of housekeeping gene elongation factor 1β (EF1β) normalized to 1.0 in uninjected embryos and statistical significance was calculated from the comparison of *SoxB2* MO-injected and *PaxA* MO-injected embryos to control MO-injected embryos.

### Squash preps and cell size analysis

Mutant cell size and abundance were measured in dissociated primary polyps (4-tentacle stage) by squash prep in *N* = 12 polyps per condition (wild type and *NvSox2* mutant). Each polyp was immobilized in 7.14% MgCl₂, fixed in 4% paraformaldehyde in PBS with 0.1% Tween (PTw) for 2 h at 4 C, washed extensively in PTw, and mounted individually in a minimal volume of glycerol on a glass slide. A glass coverslip was lowered over the fixed tissue and squashed/smeared to dissociate the fixed polyp. Mutant cell counts are presented as the number of mutant cells relative to the number of piercing cells counted at the same time in three non-overlapping visual transects through the dissociated tissue. Cell sizes were assessed in non-overlapping images captured along these transects using the Measure tool in Fiji v 1.53 f (ImageJ)[9].

### Fluorescent labeling of piercing cell harpoons

A method for labeling harpoons (the basal spiny portion of the eversible tubules in piercing cells) using Cy3-coupled tyramide was developed for this study. Primary polyps were fixed for 1 min at 25 C in 4% paraformaldehyde in PTw with 0.2% gluteraldehyde and for 1 h at 4 C in 4% paraformaldehyde in PTw. Fixed tissues were washed twice in nuclease-free water to remove PTw and stored in 100% methanol at −20C until use. For staining, tissues were slowly rehydrated from methanol into PTw and then washed three times (20 min each) in PTx (PBS with 0.2% Triton) and three times (20 min each) in 3% H₂O₂ at 25 C. Tissues were then washed three times (10 min each) in PTw to remove excess H₂O₂ and incubated in 0.2% tyramide-Cy3 (with 0.001% H₂O₂) for 45 mins in the dark. Excess tyramide and H₂O₂ were removed with three washes in PTw and polyps were counterstained with 1uM DAPI for 30 mins before being mounted on glass slides in glycerol and compressed slightly under a coverslip stabilized with clay feet. Imaging was performed the same day as labeling on a Zeiss 710 confocal. Z-stacks were rendered into 3D images using Imaris and regions of interest were demarcated using the Crop tool. Labeled harpoons from *N* = 11 polyps per condition (wild type or mutant) were counted by eye and nuclei were counted automatically using the Spots tool. Harpoon size and capsule size were measured in non-overlapping images of tentacle tips captured after laser-induced discharge of stinging cells using the Measure tool in Fiji[54].

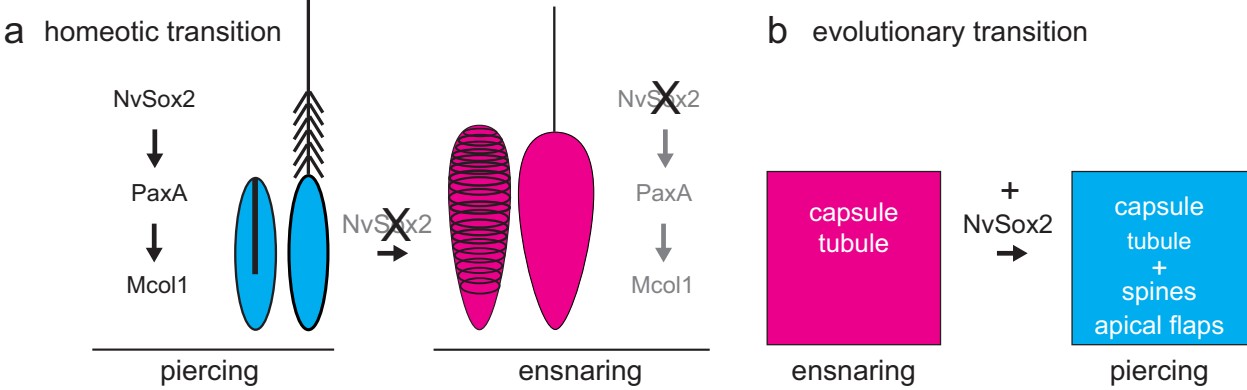

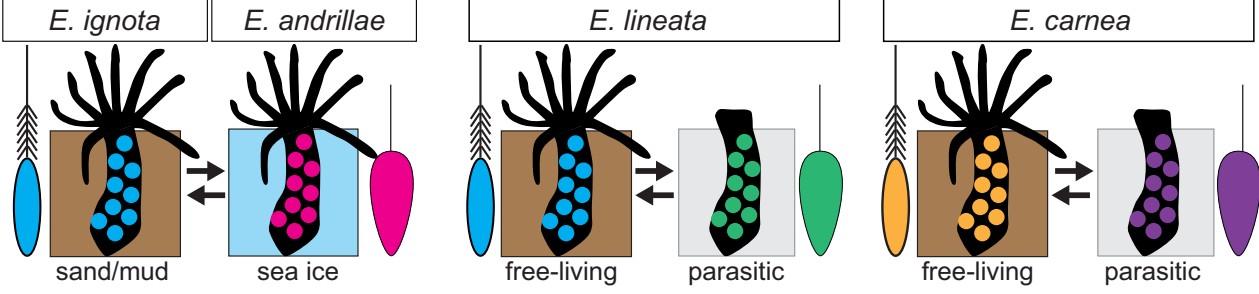

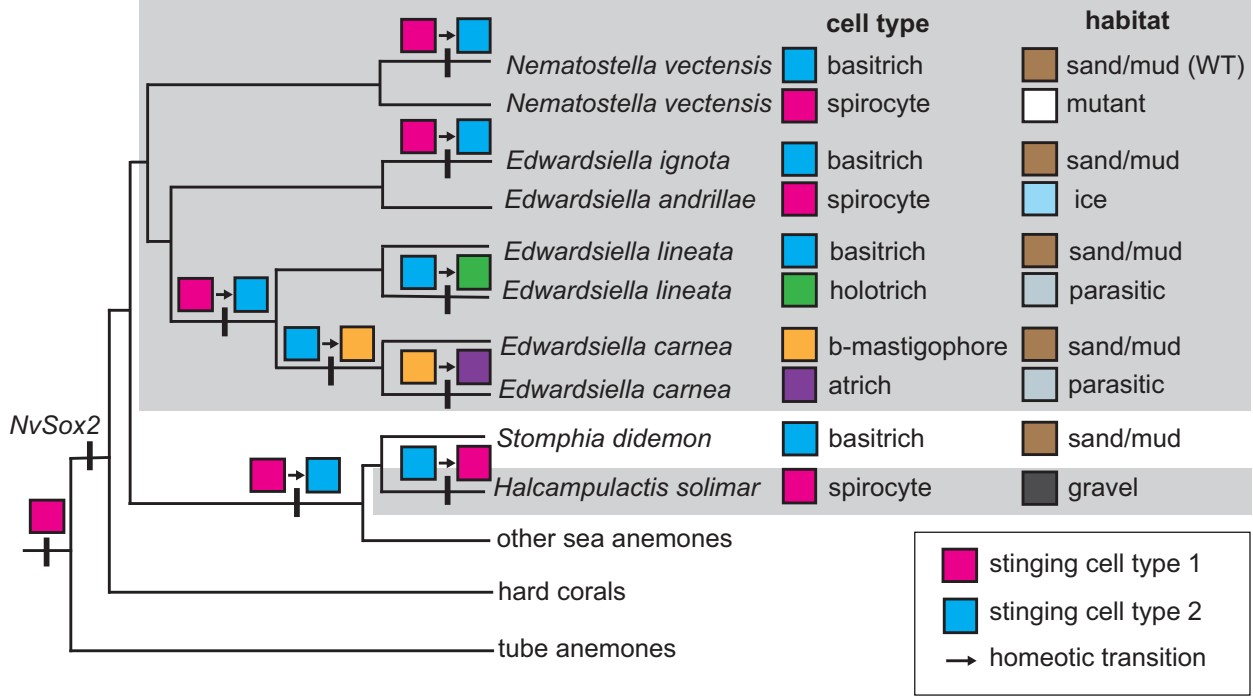

## Induction of stinging cell discharge

To induce discharge of stinging cells, juvenile polyps (6-tentacle stage) were immobilized for 5 mins in 7.14% MgCl₂ and mounted on glass slides and slightly compressed under coverslips stabilized with clay feet. An infrared laser ablation system (XY Clone, Hamilton Thorne) mounted on the 20X objective of a Zeiss Axioscope was used to induce discharge from tentacle tip stinging cells with 100% power and a 500us pulse.

## Phylogenetic analysis of *Sox* genes

Before starting any analyses, experiments were planned and described in a Phylotocol[55]. Subsequent modifications to the analyses were noted

**Fig. 6 | Homeotic transformation is a driver of stinging cell diversity. a** *NvSox2* is required for specification of small piercing cells (nematocytes, blue); ensnaring cells (spirocytes, magenta) are specified in the absence of *NvSox2*. **b** A scenario for the evolutionary co-option of an ancestral ensnaring cell with a capsule and eversible tubule to generate a piercing cell with a thick capsule wall, apical flaps, a spiny harpoon, and a tubule. **c** Sea anemones in the genus *Edwardsiella* illustrate possible roles for *NvSox2* in driving transitions between piercing cells and ensnaring cells during adaptation to different habitats. *E. ignota* and *E. andrillae* are sister taxa; *E. ignota* burrows into sand/mud and has piercing cells in its body wall and *E. andrillae* burrows into sea ice and has ensnaring cells in its body wall. *E. lineata* and *E. carnea* both shift between a free-living state (burrowing in sand/mud) and a parasitic state that lacks tentacles; this shift is mirrored by a transition between stinging cell types. Sea anemone illustrations modified from: Babonis, L. S. & Martindale, M. Q. PaxA, but not PaxC, is required for cnidocyte development in the sea anemone

*Nematostella vectensis. EvoDevo* **8**, 14 (2017) −CC BY 4.0. **d** Single-gene homeosis provides a mechanistic explanation for the wide and discontinuous variation in stinging cell types observed in the body wall of closely related cnidarians from different habitats. Ensnaring cells (magenta box) are found throughout hard corals, sea anemones, and tube anemones and likely arose in the last common ancestor of this clade. Tube anemones lack an ortholog of *NvSox2*, suggesting this gene arose in the stem ancestor of hard corals and sea anemones. Incorporation of *NvSox2* into the gene regulatory network driving ancestral ensnaring cell development in the body wall of some sea anemones (e.g., *N. vectensis* and *E. ignota*) explains why these taxa now have piercing cells (basitrichs; blue box) in the body wall whereas their close relatives (e.g., mutant *N. vectensis* and *E. andrillae*, respectively) have ensnaring cells instead. Single-gene control of cell identity could similarly drive homeotic transitions in stinging diversity throughout this clade. WT wild type.

and justified in that document. Many genes besides *Sox* genes include an HMG box, so searching for *Sox* genes using just the HMG hidden markov model (HMM) produced many non-target sequences. To identify *Sox* genes specifically, we generated a custom HMG HMM from a published *Sox* gene alignment[56] after removing the outgroup sequences (Tcf/Lef and Capicua/CIC) using hmmbuild (hmmer.org). We then used this custom HMM to search for *Sox* genes in translated transcriptomes from 15 cnidarians and six bilaterians. The abbreviations for the cnidarian taxa we used are as follows: **Aala** −*Alatina alata*, **Adig**−*Acropora digitifera*, **Amil**−*Acropora millepora*, **Epal**−*Exaiptasia pallida*, **Avan**−*Atolla vanhoeffeni*, **Ccrux**−*Calvadosia cruxmelitensis*, **Came**−*Ceriantheopsis americana*, **Chem**−*Clytia hemisphaerica*, **Cxam** −*Cassiopea xamachana*, **Elin**−*Edwardsiella lineata*, **Hech**−*Hydractinia echinata*, **Hmag**−*Hydra magnipapillata*, **Hsan**−*Haliclystus sanjuanensis*, **Nvec**−*Nematostella vectensis*, **Rren**−*Renilla reniformis*; and for bilaterians: **Bflo**−*Branchiostoma floridae*, **Cint**−*Ciona intestinalis*, **Cele** −*Caenorhabditis elegans*, **Dmel**−*Drosophila melanogaster*, **Hsap**− *Homo sapiens*, **Lgig**−*Lottia gigantea*, **Spur**−*Strongylocentrotus purpuratus*. We used this custom HMM in combination with our hmm2aln script (https://github.com/josephryan[57]) to generate an alignment that included the original sequences used to generate the HMM. We then removed all ctenophore, sponge, and placozoan sequences from this alignment and generated trees.

Phylogenetic analysis was performed following a published protocol[58]. Briefly, we used the model finder feature with IQ-TREE to identify the best substitution model for the alignment (provided as Supplementary Data 1). We then performed three maximum likelihood analyses, in parallel, using: RAxML with 25 maximum parsimony starting trees, RAxML with 25 random starting trees, and a default run

with IQ-TREE. We then compared maximum likelihood values from the outputs of all three analyses to select the best tree and performed 1000 rapid bootstraps using RAxML for branch support. The final tree file was modified in FigTree v1.4 (http://tree.bio.ed.ac.uk/software/figtree/) and Adobe Illustrator v 24.1.1 for presentation.

## Statistics and reproducibility
Pairwise comparisons of count data (Fig. 2j, l, Fig. 3n, and Fig. 5c, f) were analyzed with a 2-sided Mann−Whitney *U*-test in Microsoft Excel v 2122. Morphometric analyses of ensnaring cell size (Fig. 3l–n) were performing using ANCOVA or ANOVA with a Bonferroni post hoc analysis and qPCR data were analyzed using the delta-delta CT method and the PCR package in R (v 4.2.1)[52,53]. Electron micrographs of mature stinging cells from wild type animals (Fig. 1, Fig. 3c–f) are representative images of at least three cells of the indicated type (nematocyte, spirocyte, or ptychocyte) sampled from at least two individuals. Electron micrographs from *NvSox2* mutants (Fig. 3g–k) are representative images of at least two cells from two individuals. Exact numbers of cells imaged from each individual: Fig. 3g ($N = 6,15$), Fig. 3h ($N = 12,15$), Fig. 3i ($N = 2,2$), Fig. 3j ($N = 5,7$), Fig3k ($N = 4,5$), Fig. 4g ($N = 9,10$), Fig. 4h ($N = 5,7$), Fig. 4i ($N = 7,8$).

## Reporting summary
Further information on research design is available in the Nature Portfolio Reporting Summary linked to this article.

## Data availability
All data generated or analyzed during this study, including source data, are included in this published article (and its Supplementary

## Table 1 | Sequences for guide RNAs (gRNAs), genotyping primers for *NvSox2* mutants, qPCR primers, and morpholinos

| name | sequence (5'->3') | notes |
| --- | --- | --- |
| NvSox2_gRNA1 | TAGCGGGCGATCTATTCATTTGG | guide RNA (5'UTR) |
| NvSox2_gRNA2 | CCATGAACGCGTATATGGTATGG | guide RNA (coding sequence) |
| NvSox2_gRNA3 | CGAGTGGAATTCGCTCACTTTGG | guide RNA (coding sequence) |
| NvSox2_gRNA4 | CCAACCAATGGCTTCCCATATGG | guide RNA (coding sequence) |
| NvSox2_gRNA5 | CACCCATAGACGCACGAAGTGGG | guide RNA (coding sequence) |
| NvSox2_Cr_F primer | GGATAAGACATTCCACCCGTTTTC | forward genotyping primer |
| NvSox2_Cr_R primer | CGTTCGGTTTCTGTGGTAAAACTC | reverse genotyping primer |
| EF1B_F_primer | TGCTGCATCAGAACAGAAACCTGC | qPCR primer[22] |
| EF1B_R_primer | TAAGCCTTCAAGCGTTCTTGCCTG | qPCR primer[22] |
| NvSox2_F_primer | GGGGAGATGCCAGGGATAGGTATG | qPCR primer |
| NvSox2_R_primer | AGTGCGGGGAGAAATGGTAGGG | qPCR primer |
| control morpholino | CCTCTTACCTCAGTTACAATTTATA | standard from GeneTools |
| SoxB2 morpholino | TATACTCTCCGCTGTGTCGCTATGT | translation-blocking[22,26] |
| PaxA morpholino | AGGACCTTCAAGAACATTCGATAAT | Splice-blocking[26] |

Information files). Transgenic animals will be made available by request to the corresponding author, pending completion of a Materials Transfer Agreement. Source data are provided with this paper.

## Code availability

We used a custom script (hmm2aln) to align HMMs for phylogenetic analysis. This script is available at: https://github.com/josephryan[57].

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

## Acknowledgements

Electron microscopy was performed at the Biological Electron Microscope Facility (BEMF) at the University of Hawaiʻi, Mānoa, and at the Center for Electron Microscopy and Analysis (CEMAS), the Campus Microscopy and Imaging Facility (CMIF), and the OSU Comprehensive Cancer Center (OSUCCC) Microscopy Shared Resource (MSR) at The Ohio State University (OSU). We are grateful to Jeffrey Tonniges for his assistance with sample preparation for TEM at OSU. Funding: this work was supported by funding from the National Aeronautics and Space Administration to M.Q.M. (#NNX14AG70G), the National Science Foundation to J.F.R. (#1542597), and institutional research funds from OSU and Cornell University to M.D. and L.S.B., respectively.

## Author contributions

L.S.B., C.E., B.M.F., F.H., and M.Q.M. performed laboratory procedures including development of CRISPR reagents, microinjection, ISH, cloning and sequence analysis, and confocal microscopy. L.S.B., A.J.R., and M.D. performed electron microscopy and analyzed mutant cell phenotypes. L.S.B. and J.F.R. performed phylogenetic analyses. L.S.B. conceived of the study and wrote the manuscript. All authors edited the manuscript and approved the final draft.

## Competing interests

The authors declare no competing interests.
