## [Peer Review File · Nature Communications]

Single-cell atavism reveals an ancient mechanism of cell type diversification in a sea anemoneREVIEWER COMMENTS

Reviewer #1 (Remarks to the Author):

This is an interesting manuscript on the formation and identity of cnidocytes (stinging cells) in cnidarians. This name-giving cell type forms cnidae, highly sophisticated organelles that are released by exocytosis and release their content within fractions of a second. All cnidarians possess nematocysts which have inside the capsule a long tubule with spines that gets everted within fractions of a second and can penetrate the integument of the prey. Anthozoans have additionally spirocysts. These are capsule that also have a long tubule inside which has, however, a sticky consistency and wraps around the prey. So far, more than 20 different nematocyst-types have been described that differ in their spine, tubule, and capsule shape. The radiation of nematocytes within the Cnidaria has been regarded as a paradigm for the emergence of evolutionary novelties and cell type evolution.

There are two major weak points in the manuscript that should be considered before publication.

(i) The key experiment described in this manuscript is a CRISPR/Cas9-mediated knockout of a SOX transcription factor (NvSox2) that is expressed in nematocytes. Upon knock-out of Sox2, the morphology of nematocysts changes to that of spirocysts. This is an interesting phenotype which suggests that a whole set of genes is acting downstream of Sox2 in *Nematostella* nematocytes and silenced in spirocytes. The authors don't show such genes although they likely exist, e.g., genes encoding for spine proteins like spinalin, which have been identified in *Hydra*, but also other markers can be expected. S

Furthermore, it would be important to show that in NvSox2 mutants the staining of Minicollagen-1 is dramatically reduced, because spirocysts are lacking this minicollagen (Zenkert et al 2011). The authors describe a marker for spirocysts in *Nematostella*, the Ca²⁺-binding protein Calumenin F (Caluf). However, knockout of NvSox2 did not affect the distribution of cells expressing Caluf. Thus, it would be essential to get quantitative data on the distribution of spirocysts in WT and NvSox2 mutants. The best approach would be to perform this analysis with Minicollagen-1 and -4 antibody and determine the fraction of Minicollagen-1 negative / Minicollagen-4 (expressed in nematocysts and spirocysts) cells.

The authors also describe this effect as atavism, which would be a spectacular result, indeed. This claim is based on the fact that two types of spirocysts have been described in sea anemones, namely "robust" spirocysts with a thick wall and "gracile" spirocysts with a thin wall. *Nematostella* is described to have commonly gracile spirocysts with a thin wall, while robust spirocysts with a thick wall are common among sea anemones but are not typically found in *Nematostella vectensis*. They describe spirocyst after NvSox2 knockout to be robust with a thick wall and interpret this phenotype as an activation of an ancestral trait and an atavism. Here again the identity of the cells analyzed remains rather unclear. The phenotype of "robust" spirocysts with a thick wall and "gracile" spirocysts with a thin wall should be better documented, in wildtype *Nematostella* and in NvSox2 mutants of *Nematostella*, but also in wildtype *Halcurias* or other forms exhibiting robust spirocysts. Although robust spirocysts have been superficially described in the literature as shown in the supplemental table, I doubt that this distinction is really valid because robust spirocysts look astonishingly similar to developmental stages. Interference contrast images of the so-called robust spirocyst could just represent a developmental stage, while mature stages of spirocysts and nematocysts exhibit a more compact and thereby thinner wall. TEM picture of the capsule wall from "robust" or "gracile" spirocysts in wildtype sea anemones and NvSox2 mutants would be important to settle this unclear point. In the worst case the "robust" type of spirocysts just reflects disruptive development of spirocysts with a similar phenotype in ice sea anemones and NvSox2 mutants.

Reviewer #2 (Remarks to the Author):

This is a beautiful and compelling study from Babonis and colleagues which convincingly demonstrates the homeotic transformation of one type of cnidocyte into another by the loss of a single transcription factor in the sea anemone *Nematostella vectensis*. This result is stunning and nicely demonstrates a mechanism by which cell type diversity can arise rather quickly over the course of evolution. The study ends with a compelling argument about how diversification of cnidocytes has occurred in several closely related species of sea anemones.

A few issues should be fixed, but these issues do not undermine the data, which are quite solid:

Figure 2B qPCR: The methods say that 5 replicates were done, the panel needs errors bars

Figure 3A: Need quantification for loss of PaxA phenotype, how many animals were examined and how many had the phenotype showed in the figure?

Figure 3C: I'm having hard time figuring this panel out. I think that that the pink should represent Mcol1/Mcol4 double positive cells, not McoI1 positive only? I'm also not sure why the total in both cases isn't 100%?

Figure 3I-K: panels are missing from this figure

A few minor notes on the text:

Third line of the abstract... "these usual cells" – you might have meant "these unusual cells"?

It would have been helpful for the explanation of the different stinging cell types in *Nematostella* to come earlier, perhaps even a schematic in Figure 1. It would have helped me to clarify the results in Figure 3A,B to understand what type of stinging cells are found in the tentacle.

Reviewer #3 (Remarks to the Author):

The authors provide substantial evidence supporting the idea of a single-cell atavism in the anthozoan cnidarian, *Nematostella vectensis*, governed by the transcription factor NvSox2. First, the authors convincingly show morphological diversity of stinging cells using TEM/SEM of 3 major cnidocyte types across 3 different cnidarian species while providing a phylogenetic context for these cells and organisms. Next, the authors provide functional evidence for a change in cnidocyte composition using the NvSox2 KO. This NvSox2 KO is very clear in the blastula stage and the change in cnidocyte composition, specifically the morphology of the NvSox2 KO stinging cells compared to the WT. It is striking that the abundance of stinging cells did not change but their composition did. Further, that NvSox2 is not required for the differentiation of all stinging cell types, but in a NvSox2 KO, the cnidocyte composition is shifted from a piercing population to an ensnaring population. This work integrates functional developmental biology, a very in-depth literature review identifying cnidocytes across cnidarians, and incredible FISH/IHC/EM to support these claims. The authors also extend the idea of how these cell types may evolve by linking this functional work with known ecological data from other anemones. My major comments mostly relate to clarifying some of the writing and results shown.

Major Comments:

-Authors state that robust spirocytes are common among sea anemones but not typically found in *Nematostella*. Are there instances where they are found in *Nematostella* and can the authors clarify? - In the main text, authors discuss the transformation of stinging cells as single-cell homeosis, but if these cell types are not commonly found in *Nematostella*, the data provided support the idea of an

atavism (as the authors state), which is incredibly exciting for understanding the evolution of cell types in this informative group and phylum

-Authors claim that NvSox2 expression is not affected in response to NvPaxA knockdown in figure 2; however, it seems like NvSox2 may actually show increased expression?

-In Figure 2D, the embryos are both from the blastula stage but it is difficult to discern size differences without a scale bar. Is it possible that other structures are perturbed or there is developmental delay due to knockdown?

-Authors claim that Mcol1 expression was completely abolished in the body wall of NvSox2 mutants, but there still seems to be expression in Figure 3b

-Authors claim that the KO of NvSox2 did not affect the total number of stinging cells specified in the body wall. Were all cells counted in the body column of each individual animal?

-Figure 3D/E, authors state that small piercing cells were completely transformed into thick ensnaring cells. How do we know which cells, specifically, turned into these cells?

-In the experiments with the induced discharge of stinging cells in WT and mutant polyps, was there a measurement for the ratio of the tentacle or body length vs the harpoon shaft length? To rule out size differences between samples.

-For the induced discharge of these cell types, does the H₂O₂ tyramide only go into cells with harpoons preferentially?

-The text refers to figures Fig 3I/J/K/L but figure 3 only has up to panel H

-Authors state that KO of NvSox2 did not affect the timing of appearance or the distribution of cells expressing Caluf. Was there data for Caluf+ and Caluf+/mCol4+ in the NvSox2 KO?

Minor Comments:

-Authors state that the regulatory mechanisms controlling stinging cell diversification have been characterized which is slightly different than the developmental mechanisms driving the diversity of the stinging cell apparatus, which is what the authors are showing.

-In the summary, authors state 'usual' cells; in the rest of paper refers to them as 'unusual'

-In Figure 1 AFH, it is difficult to determine where these images are coming from, could be worth including a schematic of the organism used to highlight the region of the body they are coming from. Are they normally distributed throughout each organism presented?

-In Figure 1, At the end of the figure legend, there is a key provided for the cnidarians used in the panel. It is unclear which groups these directly correspond in the cladogram of cnidarians depicted in J. It was not immediately clear that these images were from different species until the end of the figure legend.

-Authors state that NvSox2 is one of 14 sox genes in Nematostella, one of 2 that are expressed in a pattern consistent with specification of cell identity during development, which is unclear. What are patterns consistent with specification versus other scenarios?

-Authors state that each region of Nematostella is populated by distinct combinations of stinging cell types and a schematic would be very helpful here

-In Figures 2E/F, would be helpful to know which region of the animal these are coming from. Authors state the body wall ectoderm but where in the animal... closer to oral or aboral?

-Figure 3D, is it possible that there is delayed/failed maturation of these cell types?

Response to Review

We are grateful to our three reviewers for their thoughtful consideration of this manuscript. The comments and suggestions we received have significantly improved the clarity and precision of the work. Specific responses to major and minor concerns are detailed below.

Reviewer #1 (Remarks to the Author):

This is an interesting manuscript on the formation and identity of cnidocytes (stinging cells) in cnidarians. This name-giving cell type forms cnidae, highly sophisticated organelles that are released by exocytosis and release their content within fractions of a second. All cnidarians possess nematocysts which have inside the capsule a long tubule with spines that gets everted within fractions of a second and can penetrate the integument of the prey. Anthozoans have additionally spirocysts. These are capsule that also have a long tubule inside which has, however, a sticky consistency and wraps around the prey. So far, more than 20 different nematocyst-types have been described that differ in their spine, tubule, and capsule shape. The radiation of nematocytes within the Cnidaria has been regarded as a paradigm for the emergence of evolutionary novelties and cell type evolution.

There are two major weak points in the manuscript that should be considered before publication.

(i) The key experiment described in this manuscript is a CRISPR/Cas9-mediated knockout of a SOX transcription factor (NvSox2) that is expressed in nematocytes. Upon knock-out of Sox2, the morphology of nematocysts changes to that of spirocysts. This is an interesting phenotype which suggests that a whole set of genes is acting downstream of Sox2 in *Nematostella* nematocytes and silenced in spirocytes. The authors don't show such genes although they likely exist, e.g., genes encoding for spine proteins like spinalin, which have been identified in *Hydra*, but also other markers can be expected.

Response: *This is a logical suggestion and we are indeed very interested in identifying the downstream targets of NvSox2 in a subsequent study. The results we present in this manuscript show there are two different (and opposite) effects of NvSox2 knockout on the harpoon in two different lineages of piercing cells (small and large basitrichous isorhiza nematocytes). In the small piercing cells found throughout the body wall, NvSox2 knockout resulted in a complete loss of the harpoon (Fig 3G). By contrast, in the large piercing cells found in the tentacle tips, knockout of NvSox2 resulted in elongation of the harpoon (Fig 5E,F). Because of these opposing effects, qPCR performed in whole embryos (wild types and mutants) will likely reveal no difference in the quantitative expression of the genes that encode the harpoon (including the spinalin-like gene reported by Karabulut et al, 2022). Thus the only way to assess the unique effects of NvSox2 on harpoon-specific genes like spinalin-like is to perform single-cell analyses (scRNA-seq and ChIP-seq), which we have planned for the next step in this line of research. To address this concern, we have added language to show that these combined observations have motivated our next steps in this line of research, as follows:*

*Lines 296-307: "The fact that NvSox2 controls only the length of the harpoon in large piercing cells suggests it should be possible to find individual genes that similarly control the morphology of the spines, the number of barbs, the composition of the capsule wall, and the shape of the apical structures. Indeed, a recent study of piercing cell function in *N. vectensis* identified a spinalin-like protein⁴¹ as a specific component of the harpoon⁴². Future studies investigating the regulatory relationships of transcription factors, like NvSox2, and effector genes driving harpoon morphology, including spinalin-like, on a single-cell basis will provide a unique opportunity to reconstruct the evolutionary diversification of each individual stinging cell type and lead to a deeper understanding of how gene regulatory networks evolve. Similar to the modular control of neural identity described from nematodes⁴³, this framework explains how evolution could mix-and-match genes controlling fine aspects of subcellular phenotype to produce the diverse array of cell types that drive animal biodiversity and provides a model for characterizing the evolution of other novel organelles."*

(ii) Furthermore, it would be important to show that in NvSox2 mutants the staining of Minicollagen-1 is dramatically reduced, because spirocysts are lacking this minicollagen (Zenkert et al 2011). The authors describe a marker for spirocysts in *Nematostella*, the Ca²⁺-binding protein Calumenin F (Caluf). However, knockout of NvSox2 did not affect the distribution of cells expressing Caluf. Thus, it would be

essential to get quantitative data on the distribution of spirocysts in WT and NvSox2 mutants. The best approach would be to perform this analysis with Minicollagen-1 and -4 antibody and determine the fraction of Minicollagen-1 negative / Minicollagen-4 (expressed in nematocysts and spirocysts) cells.

Response: *This has been fixed, as suggested. The co-expression of Mcol4 and Mcol1 is shown in Fig 2K using fluorescent ISH/IHC for wild type and NvSox2 mutant animals. Additionally, we have provided Figure 2L to demonstrate the quantitative change in the proportion of cells co-expressing Mcol1 and Mcol4 in the body wall of N. vectensis. These changes are described in the text and figure caption as indicated below.*

Lines 115-119: *“Quantitative analysis of stinging cell development revealed that 80% of the Mcol4-labeled cells in the body wall of wild type polyps also express Mcol1 and this proportion is significantly reduced to less than 10% in NvSox2 mutants (Fig 2L). These results confirm that knockout of NvSox2 did not affect the total number of stinging cells specified in the body wall but transformed the identity of the stinging cells in this tissue.”*

Caption Fig 2, Lines 512-514: *“(L) The percent of stinging cells in the body wall co-expressing Mcol1 and Mcol4 is significantly lower in NvSox2 mutants than wildtype animals ($p = 1.57E-4$). N = 10 animals per treatment (tentacle bud stage).”*

(iii) The authors also describe this effect as atavism, which would be a spectacular result, indeed. This claim is based on the fact that two types of spirocysts have been described in sea anemones, namely “robust” spirocysts with a thick wall and “gracile” spirocysts with a thin wall. Nematostella is described to have commonly gracile spirocysts with a thin wall, while robust spirocysts with a thick wall are common among sea anemones but are not typically found in Nematostella vectensis. They describe spirocyst after NvSox2 knockout to be robust with a thick wall and interpret this phenotype as an activation of an ancestral trait and an atavism. Here again the identity of the cells analyzed remains rather unclear. The phenotype of “robust” spirocysts with a thick wall and “gracile” spirocysts with a thin wall should be better documented, in wildtype Nematostella and in NvSox2 mutants of Nematostella, but also in wildtype Halcurias or other forms exhibiting robust spirocysts. Although robust spirocysts have been superficially described in the literature as shown in the supplemental table, I doubt that this distinction is really valid because robust spirocysts look astonishingly similar to developmental stages. Interference contrast images of the so-called robust spirocyst could just represent a developmental stage, while mature stages of spirocysts and nematocysts exhibit a more compact and thereby thinner wall. TEM picture of the capsule wall from “robust” of “gracile” spirocysts in wildtype sea anemones and NvSox2 mutants would be important to settle this unclear point. In the worst case the “robust” type of spirocysts just reflects disruptive development of spirocysts with a similar phenotype in ice sea anemones and NvSox2 mutants.

Response: *To address the confusion over stinging cell morphology, we have provided clarifying language in several places (as indicated below) regarding the morphology of these different cell types and included new SEMs (Fig 1G-I) further highlighting the differences in the capsule morphology of nematocytes and spirocytes. As suggested, we have also provided TEMs of gracile spirocytes from both wildtype (Fig 4D-F) and mutant animals (Fig 4G-I) showing no differences. These changes are indicated in the text and figure caption as follows:*

Lines 51-63: *“Three major subtypes of stinging cells are currently recognized and can be discriminated by the morphology of their stinging organelle¹¹. The first, nematocytes (“piercing cells”) (Fig 1A-C,G-H), have an explosive organelle with a thick capsule wall containing an eversible, venom-laden tubule¹². Across cnidarians, over 30 different types of nematocytes have been described, differing largely in the morphology of the basal, spiny portion of the tubule (hereafter called the “harpoon”) ⁸. The second stinging cell type, spirocytes (“ensnaring cells”), exhibit much less morphological variation and are described as either gracile (thin) or robust (thick) in overall appearance¹³. While still extrusive, these cells are typified by two traits not found in piercing cells: a thin capsule wall with a serrated appearance, derived from a network of regularly spaced fibers lining the inner capsule wall, and an eversible tubule adorned with fine lateral rods that create an ensnaring web upon discharge¹⁴ (Fig 1D-F,I). The third group of stinging cells, ptychocytes (“adherent cells”), are not known to contain toxins and their payload consists of only a pleated, sticky tubule folded inside a thin-walled capsule¹⁵ (Fig 1J-L).”*

Caption Fig 1, Lines: 473-477: *“(E,F) Undischarged spirocytes (TEM) from N. vectensis showing an apical cap (orange, false colored) and a thin, serrated capsule wall (white arrowheads). The serrated*

appearance of the spirocyte capsule wall arises from an internal network of regularly spaced fibers (white arrowheads in F). Fine lateral rods adorn the tubule and appear as small, dark puncta in cross section (black arrowhead in E)."

Lines 129-136: "Stinging cells from the body wall of wild type polyps had rigid capsules with a thick capsule wall and prominent flaps at the apical (outward facing) end (**Fig 3C-E**). When discharged, these stinging cells revealed an extrusive tubule with large spines proximally (the "harpoon") and small barbs distally (**Fig 3F**). These features are diagnostic of sea anemone piercing cells^{22,26,27}. By contrast, the capsule of the extruded mutant cells appeared flimsy, collapsing upon discharge, and the extrusive apparatus consisted of a smooth tubule devoid of spines and barbs (**Fig 3G**). The mutant cells also have a thin capsule wall with a serrated appearance along the inner surface and a flat apical cap (**Fig 3H-J**), features consistent with ensnaring cells, not piercing cells.^{14,28,29}"

Regarding the possibility that the mutant cells are simply immature/underdeveloped, we have provided TEMs showing that the mutant cells have all the features of a mature spirocyte (apical cap, serrated capsule wall, and internalized tubule; Fig 3H-J) and provided a new figure in the supplement (Supplementary Fig. 4) showing the anti-Mcol4 antibody only detects immature cells (as previously described by Zenkert et al., 2011). To further reduce confusion, we have now explicitly labeled the stage of animals in all figures demonstrating that mature mutant cells were detected at multiple life stages, from tentacle bud (Fig 2,5) to juvenile polyp (6-tentacle stage; Fig 3,5). Finally, if these mutant cells were immature we would not have been able to induce their discharge in preparation for SEM analysis (Fig 3G). Together, we think it is reasonable to conclude that the mutant cells are fully differentiated spirocytes but are not the same type of spirocyte found in the tentacle tip. Supplementary Fig 4 is referenced in the text as follows. The text of the caption is also provided below.

Lines 114-115: "By contrast, minicollagen-4 (Mcol4), which labels all types of stinging cells before maturation of the capsule (**Supplementary Fig. 4**), was unaffected."

Caption Supplementary Fig 4: "Detection of immature stinging cells with α -Mcol4 antibody in NvSox2 mutant animals. (A) α -Mcol4 labeled mutant stinging cells in the body wall of a tentacle bud stage animal shown in a 3D rendered z-stack (max projection) and an individual optical section (z-plane). Cells labeled with α -Mcol4 are immature and do not yet have a capsule that is visible in DIC (white circle) as previously shown for wild type *N. vectensis*^{3,4}. Ecto – ectoderm, M – mesoglea. (B) α -Mcol4 labels immature stinging cells in the tentacle tip of a tentacle bud stage animal. A developing large piercing cell (basitrichous isorhiza nematocyte) is labeled with α -Mcol4 (white arrowhead); at this stage the capsule is not yet visible with DIC (white circle). A mature mutant cell (orange arrowheads) has a capsule that can clearly be seen in DIC (false colored orange) but it not labeled with α -Mcol4 antibody."

To address the problem that the definition of "gracile" and "robust" spirocytes provided in the literature is rather unclear, we have analyzed images of "gracile" and "robust" spirocytes from the literature (Source Data) to identify quantitative features to discriminate these two cell types. The results of this analysis are provided in Fig 3L-M) and are described as indicated below.

Lines 147-157: "Despite having the canonical morphological features of mature ensnaring cells, the cells that developed in the body wall of NvSox2 mutants were morphologically distinct from the gracile ensnaring cells that develop in the tentacle tips of wild type animals, leading us to hypothesize that loss of NvSox2 caused the development of robust ensnaring cells in the body wall of *N. vectensis*. The "robust" and "gracile" conditions are not well-defined; to remedy this, we analyzed the length and width of the capsule from undischarged ensnaring cells from the literature reported to be either "gracile" or "robust". The relationship between capsule width and length did not differ for gracile and robust ensnaring cells across a range of sizes (Fig 3L); however, for a given length, robust ensnaring cells were significantly wider than gracile cells (Fig 3L,M). The width of the capsule that developed in the body wall of NvSox2 mutants is distinct from and intermediate between the capsule width reported for gracile and robust ensnaring cells in the literature (Fig 3M)."

Reviewer #2 (Remarks to the Author):

This is a beautiful and compelling study from Babonis and colleagues which convincingly demonstrates the homeotic transformation of one type of cnidocyte into another by the loss of a single transcription factor in the sea anemone *Nematostella vectensis*. This result is stunning and nicely demonstrates a

mechanism by which cell type diversity can arise rather quickly over the course of evolution. The study ends with a compelling argument about how diversification of cnidocytes has occurred in several closely related species of sea anemones.

A few issues should be fixed, but these issues do not undermine the data, which are quite solid:

Figure 2B qPCR: The methods say that 5 replicates were done, the panel needs errors bars

Response: We have updated this figure (now Figure 2F) to explicitly show the results of each replicate experiment (black dots) on top of the average for the treatment (grey bars) and added error bars and p-values (in caption). The text of the caption has been updated as follows:

Caption Fig 2, Lines: 499-503: "(F) qPCR shows *NvSox2* is significantly downregulated in response to *SoxB2* knockdown ($p = 1.12E-2$) but not significantly affected by knockdown of *PaxA* ($p = 0.345957$). Data are presented as fold-change relative to uninjected controls (normalized to 1.0). Results of three replicate experiments for each treatment are shown (black dots) on top of averages of all three experiments (grey bars)."

Figure 3A: Need quantification for loss of PaxA phenotype, how many animals were examined and how many had the phenotype showed in the figure?

Response: These data are now presented in Figure 2. We have added Fig 2J quantifying the number of PaxA-expressing cells found in the two treatments. Sample sizes are now indicated in the figure caption, which has been updated to read as follows:

Caption Fig 2, Lines 507-509: "(J) The number of cells in the body wall expressing *PaxA* is significantly lower in *NvSox2* mutants than wildtype (WT) animals ($p = 1.37E-11$). $N = 32$ (WT) and $N = 30$ (mutant) animals were examined per treatment (tentacle bud stage)."

Figure 3C: I'm having hard time figuring this panel out. I think that that the pink should represent Mcol1/Mcol4 double positive cells, not Mcol1 positive only? I'm also not sure why the total in both cases isn't 100%?

Response: This figure (now Fig 2L) has been updated to show only the number of cells co-expressing Mcol1 and Mcol4 in both groups (wildtype and *NvSox2* mutants). These changes have been described in the text and figure caption as follows:

Lines 115-119: "Quantitative analysis of stinging cell development revealed that 80% of the Mcol4-labeled cells in the body wall of wild type polyps also express Mcol1 and this proportion is significantly reduced to less than 10% in *NvSox2* mutants (Fig 2L). These results confirm that knockout of *NvSox2* did not affect the total number of stinging cells specified in the body wall but transformed the identity of the stinging cells in this tissue."

Caption Fig 2, Lines 512-514: "(L) The percent of stinging cells in the body wall co-expressing Mcol1 and Mcol4 is significantly lower in *NvSox2* mutants than wildtype animals ($p = 1.57E-4$). $N = 10$ animals per treatment (tentacle bud stage)."

Figure 3I-K: panels are missing from this figure

Response: This has been fixed.

A few minor notes on the text:

Third line of the abstract... "these usual cells" – you might have meant "these unusual cells"?

Response: This has been fixed.

It would have been helpful for the explanation of the different stinging cell types in *Nematostella* to come earlier, perhaps even a schematic in Figure 1. It would have helped me to clarify the results in Figure 3A,B to understand what type of stinging cells are found in the tentacle.

Response: To address this concern, we have added a schematic to the top of Fig 2 (panels A-D) that summarizes the distribution of the different stinging cell types in *N. vectensis*. These changes are described in the text as follows:

Lines 82-89: "In *N. vectensis*, stinging cells are found throughout the ectodermal epithelia but each region of the animal is populated by a distinct combination of stinging cell types²². The tentacle tips, for example, are populated by large and small basitrichous isorhiza nematocytes (piercing cells) and gracile

spirocytes (ensnaring cells) whereas the body wall (outer epithelium) is populated exclusively by large and small piercing cells (**Fig. 2A,B**). The ectodermal epithelia of the mesenteries (digestive tissues) are populated largely by microbasic mastigophores (another type of piercing cell) and the foot is populated almost exclusively by small basitrichous isorhizas (**Fig. 2C,D**). The distributions and relative abundance of the different stinging cell types are summarized in Fig. 2B.”

Caption Fig 2, Lines 491-496: “(A-D) Stinging cells are distributed throughout the ectoderm in *Nematostella vectensis*. (A) Juvenile polyp labeled with 143uM DAPI showing nuclei and mature stinging cell capsules; small (blue) and large (yellow) basitrichs are indicated. DIC microscopy (inset) shows abundant spirocytes (magenta) in the tentacles. (B) Summary of the distribution of stinging cell types in *N. vectensis*. (C) Mesenteries are populated largely by mastigophores (green); (D) small basitrichs dominate the body wall and foot (labeled with DAPI).”

Reviewer #3 (Remarks to the Author):

The authors provide substantial evidence supporting the idea of a single-cell atavism in the anthozoan cnidarian, *Nematostella vectensis*, governed by the transcription factor NvSox2. First, the authors convincingly show morphological diversity of stinging cells using TEM/SEM of 3 major cnidocyte types across 3 different cnidarian species while providing a phylogenetic context for these cells and organisms. Next, the authors provide functional evidence for a change in cnidocyte composition using the NvSox2 KO. This NvSox2 KO is very clear in the blastula stage and the change in cnidocyte composition, specifically the morphology of the NvSox2 KO stinging cells compared to the WT. It is striking that the abundance of stinging cells did not change but their composition did. Further, that NvSox2 is not required for the differentiation of all stinging cell types, but in a NvSox2 KO, the cnidocyte composition is shifted from a piercing population to an ensnaring population. This work integrates functional developmental biology, a very in-depth literature review identifying cnidocytes across cnidarians, and incredible FISH/IHC/EM to support these claims. The authors also extend the idea of how these cell types may evolve by linking this functional work with known ecological data from other anemones. My major comments mostly relate to clarifying some of the writing and results shown.

Major Comments:

-Authors state that robust spirocytes are common among sea anemones but not typically found in *Nematostella*. Are there instances where they are found in *Nematostella* and can the authors clarify?
Response: To clarify this point, we have provided Fig 3N showing the abundance of cells with the mutant morphology in wild type and NvSox2 mutant animals. These changes are discussed in the text and figure caption as follows:

Lines 157-166: “Surprisingly, examination of wild type animals revealed the presence of cells with this mutant morphology, albeit in very low frequency. While the mutant cell type comprised nearly 50% of the stinging cells in NvSox2 mutants, fewer than 1% of the stinging cells of wild types had this morphology (**Fig 3N**). Of the 12 wild type polyps examined, only one appeared to lack these mutant cells completely. These results suggest errors may arise spontaneously in the NvSox2 regulatory pathway at low frequency resulting in the development of robust ensnaring cells in wild type animals. Because robust ensnaring cells are common and abundant in other sea anemones (Source Data), it is reasonable to suggest that this cell type was likely present in the most recent common ancestor of all sea anemones. Therefore, our results suggest knockout of NvSox2 has resulted in single-cell atavism, or the restoration of an ancestral cell type (robust ensnaring cell) in *N. vectensis*.”

Caption Fig 3, Lines 537-538: “(N) Cells with the mutant phenotype arise spontaneously but are significantly rarer in wild types than in mutant polyps ($p = 3.2E-5$); $N = 12$ polyps per treatment (4-tentacle stage).”

-In the main text, authors discuss the transformation of stinging cells as single-cell homeosis, but if these cell types are not commonly found in *Nematostella*, the data provided support the idea of an atavism (as the authors state), which is incredibly exciting for understanding the evolution of cell types in this informative group and phylum

Response: We clarify this point by defining the type of atavism demonstrated in this study as a special case of homeosis and by explicitly describing how we infer cell type ancestry from the evolutionary relationship of sea anemones. These changes are indicated in the text as follows:

Lines 162-166: "Because robust ensnaring cells are common and abundant in other sea anemones (Source Data), it is reasonable to suggest that this cell type was likely present in the most recent common ancestor of all sea anemones. Therefore, our results suggest knockout of *NvSox2* has resulted in single-cell atavism, or the restoration of an ancestral cell type (robust ensnaring cell) in *N. vectensis*."

Lines 273-280: "Together, these results confirm that this mutation is functionally relevant and is, therefore, a bona fide homeotic transformation⁷. Importantly, the transformation resulted in restoration of a type of ensnaring cell that was lost in the ancestor of the lineage that gave rise to *Nematostella vectensis*. Thus, we have shown a special case of homeosis: single-cell atavism, or the restoration of an ancestral cell type in a modern species. As this is the first evidence of atavism in a cnidarian, our results suggest that homeotic control of cell fate is an ancient mechanism that has contributed to the expansion of biodiversity since the last common ancestor of cnidarians and bilaterians, over 800MYA³⁷."

-Authors claim that *NvSox2* expression is not affected in response to *NvPaxA* knockdown in figure 2; however, it seems like *NvSox2* may actually show increased expression?

Response: We have provided p-values for these analyses and clarified the language used to describe these results as indicated below in the text and figure caption.

Lines 97-100: "Using qPCR in combination with morpholino-mediated gene knockdown, we found significant suppression of *NvSox2* expression following knockdown of *SoxB2* (expressed in progenitors of neurons and stinging cells)¹⁹ and no significant change in *NvSox2* expression in response to knockdown of *PaxA* (Fig 2F)."

Caption Fig 2, Lines 499-500: "(F) qPCR shows *NvSox2* is significantly downregulated in response to *SoxB2* knockdown ($p = 1.12E-2$) but not significantly affected by knockdown of *PaxA* ($p = 0.345957$)."

-In Figure 2D, the embryos are both from the blastula stage but it is difficult to discern size differences without a scale bar. Is it possible that other structures are perturbed or there is developmental delay due to knockdown?

Response: To address the concern about embryo size, we have added scale bars to all the images in our figures. To address the potential for developmental delay, we have provided a side-by-side comparison of early development in wild type and *NvSox2* mutant embryos labeled with *NvSox2* probe in Supplementary Figure 2. These images do not show any obvious developmental delays as key developmental landmarks appear at the same time in both treatments.

-Authors claim that *Mcol1* expression was completely abolished in the body wall of *NvSox2* mutants, but there still seems to be expression in Figure 3b

Response: To address this point we have now provided a summary of the distribution of stinging cell types in *N. vectensis* (Fig 2A-D) indicating that small piercing cells are dominant in the body wall but are not the only stinging cell type specified in this tissue. We have modified the language in the text to reflect this detail as indicated below.

Lines 112-114: "Expression of the transcription factor *PaxA* (Fig 2I,J; Supplementary Fig 3) and the structural molecule *minicollagen-1* (*Mcol1*; Fig 2K,L) (both piercing cell-specific) was significantly reduced in the body wall of *NvSox2* mutants."

-Authors claim that the KO of *NvSox2* did not affect the total number of stinging cells specified in the body wall. Were all cells counted in the body column of each individual animal?

Response: We have provided additional detail in the methods section of the Supplementary Information to clarify this point, as indicated below.

Supplementary Information, Methods: "**Analysis of *Mcol4* and *Mcol1* expression:** Developing animals were fixed for examination at the tentacle bud stage (240 hours post fertilization). We performed a combination of *in situ* hybridization and immunohistochemistry on the same tissues as previously described³. Labeled tissues were then mounted in glycerol on glass slides, imaged with a Zeiss 710 confocal microscope, and z-stacks were rendered into 3D images using Imaris software (Oxford Instruments, USA). Regions of interest were demarcated using the crop tool in Imaris and labeled cells were counted by eye in a 100 x 100um square region of interest aboral to the tentacle buds. We

compared the number of cells labeled with Mcol4 in wild type and NvSox2 mutants using a Mann Whitney U test and found no significant difference in the total number of Mcol4-labeled cells ($p = 0.623176$). We then counted the number of cells in the same region of interest that co-expressed Mcol1 and Mcol4 and found a significant decrease in NvSox2 mutants, relative to wild type animals ($p = 1.57E-4$). Data are presented (Fig 2L) as percent of cells labeled with Mcol4 antibody that were also labeled with Mcol1 mRNA probe. $N = 10$ tentacle bud stage animals were examined in each treatment.”

-Figure 3D/E, authors state that small piercing cells were completely transformed into thick ensnaring cells. How do we know which cells, specifically, turned into these cells?

Response: We have added language to address this uncertainty as follows:

Lines 139-144: “Without cell lineage labeling, we cannot rule out the possibility that knockout of NvSox2 caused both the loss of piercing cells and the gain of ensnaring cells in two distinct cell lineages in the body wall of *N. vectensis*. Considering loss of NvSox2 did not affect the number of stinging cells expressing Mcol4 in the body wall, the simplest explanation for these results is that loss of NvSox2 causes a transformation of small piercing cells into ensnaring cells in the body wall of *N. vectensis*.”

-In the experiments with the induced discharge of stinging cells in WT and mutant polyps, was there a measurement for the ratio of the tentacle or body length vs the harpoon shaft length? To rule out size differences between samples.

Response: To clarify this, we have provided Fig 5E-F showing harpoon length relative to capsule length for wild type and NvSox2 mutant animals. These changes are indicated in the text, the figure caption, and the supplementary methods as follows:

Lines 205-209: “Examination of the discharged piercing cells suggested that the looped phenotype of the mutants is due to elongation of the harpoon (Fig 5D-E). To quantify this effect, we measured the length of the harpoon and the length of the capsule in discharged stinging cells from the tentacle tips of wild type and mutant polyps and show a significant increase (~50%) in the length of the harpoon in mutant animals (Fig 5F).”

Caption Fig 5, Lines 565-566: (F) Discharged harpoons are significantly longer in large piercing cells from mutant animals than wild types ($p = 2.1E-13$). Number of cells analyzed: $N = 57$ (WT), $N = 41$ (mutant).”

Supplementary Information, Methods: “Harpoon size and capsule size were measured in non-overlapping images of tentacle tips captured after laser-induced discharge of stinging cells using the Measure tool in Fiji⁸. Data are presented (Fig 5F) as harpoon length / capsule length and the two groups (wild type and NvSox2 mutant) were compared with a Mann Whitney U test. Number of cells analyzed: $N = 57$ (WT), $N = 41$ (NvSox2 mutant).”

-For the induced discharge of these cell types, does the H₂O₂ tyramide only go into cells with harpoons preferentially?

Response: We have added Figs 2A-B, 4A, and 5A to show more clearly that the tentacles are populated by numerous gracile spirocytes and large nematocytes but the images in Fig 5B show only the nematocytes are labeled with H₂O₂. To further clarify this point, we have added the following text to the methods section:

Lines 320-322: “A method for labeling harpoons (the basal spiny portion of the eversible tubules in piercing cells) using H₂O₂-Cy3 tyramide was developed for this study.”

-The text refers to figures Fig 3I/J/K/L but figure 3 only has up to panel H

Response: This has been fixed.

-Authors state that KO of NvSox2 did not affect the timing of appearance or the distribution of cells expressing Caluf. Was there data for Caluf+ and Caluf+/mCol4+ in the NvSox2 KO?

Response: We have added Fig 4C showing the expression of CaluF (by colorimetric ISH) does not differ in wild type and NvSox2 mutant animals throughout early development and Fig 4D-I showing that there are no morphological differences between the gracile spirocytes in wild type and mutant polyps. These changes are described in the text as follows:

Lines 179-186: “Knockout of NvSox2 did not affect the timing of appearance or the distribution of cells expressing CaluF (Fig 4C), confirming the mutant ensnaring cells are not gracile cells. Furthermore, analysis of gracile cell morphology by TEM confirmed that gracile cells from wild type and mutant animals

are morphologically indistinguishable (**Fig 4D-I**). In light of these results, we suggest that knockout of *NvSox2* causes a homeotic transformation of small piercing cells into robust ensnaring cells anywhere small piercing cells would normally be specified. By contrast, loss of *NvSox2* did not affect the development of gracile ensnaring cells, suggesting gracile and robust ensnaring cells are patterned using distinct mechanisms.”

Caption Fig 4, Lines 547-551: “(C) Expression of *CaluF* mRNA in wild type (WT) and mutant embryos showing no effect of *NvSox2* knockout on the onset of expression or the distribution of *CaluF*-expressing cells. (D-I) TEMs of gracile spirocytes from wild type and mutant animals. Knockout of *NvSox2* did not affect the morphology of the serrated capsule wall (white arrowheads) or the lateral rods on the internalized tubule (black arrowheads).”

Minor Comments:

-Authors state that the regulatory mechanisms controlling stinging cell diversification have not been characterized which is slightly different that the developmental mechanisms driving the diversity of the stinging cell apparatus, which is what the authors are showing.

Response: *This has been fixed as suggested.*

Lines 66-69: “However, the developmental mechanisms controlling variability in the morphology of the stinging apparatus have not been characterized. The evolutionary factors driving the diversification of these unusual cell types, therefore, remains unresolved.”

-In the summary, authors state ‘usual’ cells; in the rest of paper refers to them as ‘unusual’

Response: *This has been fixed as suggested.*

Lines 20-22: “These unusual cells are iconic examples of biological novelty but the developmental mechanisms driving diversity of the stinging apparatus are poorly characterized, making it challenging to understand their evolutionary history.”

-In Figure 1 AFH, it is difficult to determine where these images are coming from, could be worth including a schematic of the organism used to highlight the region of the body they are coming from. Are they normally distributed throughout each organism presented?

Response: *We have updated the caption for Fig 1 to include the part of the animal from which these cells were collected as follows:*

Caption Fig 1, Lines 468-484: “(A,B) Discharged nematocyte (SEM) from the mesentery of the sea anemone *Nematostella vectensis* showing apical flaps (B - green, false colored) and spines along the everted harpoon (white arrow). (C) Apex of undischarged nematocyte (TEM) from the mesentery of *N. vectensis* showing apical flaps (green) and thick capsule wall (arrowheads). (D) Discharged spirocytes (SEM) from the tentacles of the sea anemone *Calliactis tricolor* showing lack of apical flaps and no spines on the everted tubule (white arrow). (E,F) Undischarged spirocytes (TEM) from the tentacles of *N. vectensis* showing an apical cap (orange, false colored) and a thin, serrated capsule wall (white arrowheads). The serrated appearance of the spirocyte capsule wall arises from an internal network of regularly spaced fibers (white arrowheads in F). Fine lateral rods adorn the tubule and appear as small, dark puncta in cross section (black arrowhead in E). (G,H) SEMs of a broken nematocyst capsule from *N. vectensis* showing the thick capsule wall (black arrowheads). (I) Intact spirocyte capsule from *N. vectensis*; the coils of the tubule are visible through the thin capsule wall (two coils are delineated with dashed lines). (J) Discharged ptychocyte (SEM) from the body wall of the tube anemone *Ceriantheopsis americana*; the everted tubule lacks spines but has longitudinal pleats (white arrow). (K) Apex of undischarged ptychocyte from the body wall of *C. americana* showing no specialization (purple, false colored). (L) Cross section of pleated tubule inside the capsule of an undischarged ptychocyte from *C. americana*; the capsule wall is thin and not serrated (white arrowheads).”

-In Figure 1, At the end of the figure legend, there is a key provided for the cnidarians used in the panel. It is unclear which groups these directly correspond in the cladogram of cnidarians depicted in J. It was not immediately clear that these images were from different species until the end of the figure legend.

Response: *We have updated the caption for Fig 1 to explicitly list species names (and clade) for each image, as indicated in the previous response.*

-Authors state that NvSox2 is one of 14 sox genes in Nematostella, one of 2 that are expressed in a pattern consistent with specification of cell identity during development, which is unclear. What are patterns consistent with specification versus other scenarios?

Response: To reduce confusion, we have modified this language as follows:

Lines 92-94: "NvSox2 is one of 14 Sox genes in *N. vectensis*, only two of which (SoxB2 and NvSox2) are expressed in individual cells throughout the ectoderm, a pattern consistent with specification of terminal cell identity during early embryogenesis ¹⁷."

-Authors state that each region of Nematostella is populated by distinct combinations of stinging cell types and a schematic would be very helpful here

Response: We have added Fig 2A-D to remedy this confusion. These changes are described in the text and caption as follows:

Lines 82-89: "Stinging cells are found throughout the ectodermal epithelia in juvenile polyps but each region of the animal is populated by a distinct combination of stinging cell types ²². The tentacle tips are populated by large and small basitrichous isorhiza nematocytes (piercing cells) and gracile spirocytes (ensnaring cells) while the body wall (outer epithelium) is populated exclusively by large and small piercing cells (**Fig 2A,B**). The ectodermal epithelia of the mesenteries (digestive tissues) are populated largely by microbasic mastigophores (another type of piercing cell) and the foot is populated exclusively by small basitrichous isorhizas (**Fig 2C,D**). The distributions and relative abundance of the different stinging cell types are summarized in Fig 2B."

Caption Fig 2, Lines 491-496: "(A-D) Stinging cells are distributed throughout the ectoderm in *Nematostella vectensis*. (A) Juvenile polyp labeled with 143uM DAPI showing nuclei and mature stinging cell capsules; small (blue) and large (yellow) basitrichs are indicated. DIC microscopy (inset) shows abundant spirocytes (magenta) in the tentacles. (B) Summary of the distribution of stinging cell types in *N. vectensis*. (C) Mesenteries are populated largely by mastigophores (green); (D) small basitrichs dominate the body wall and foot (labeled with DAPI)."

-In Figures 2E/F, would be helpful to know which region of the animal these are coming from. Authors state the body wall ectoderm but where in the animal... closer to oral or aboral?

Response: These panels are now shown in Fig 3B and we have included Fig 3A (schematic) to highlight the region of the animal from which the data in panel B were collected. These changes are reflected in the figure caption as follows:

Caption Fig 3, Lines : "(A) NvSox2 is required for expression of PaxA and Mcol1 in the body wall, which is dominated by small piercing cells (blue). The blue box also indicates the region of the animal examined in panel B. (B) The small piercing cells (blue arrowheads) in the body wall ectoderm (Ecto) of wild type (WT) polyps are replaced by mutant cells (white arrowheads) in NvSox2 mutants. Section: 1um thick sections counterstained with toluidine blue (M – mesoglea); surface: DIC imaging, cells are false colored."

-Figure 3D, is it possible that there is delayed/failed maturation of these cell types?

Response: We have addressed this concern as indicated above (to Reviewer 1). To summarize, we have provided TEMs showing that the mutant cells have all the features of a mature spirocyte (apical cap, serrated capsule wall, and internalized tubule; Fig 3H-J) and provided a new figure in the supplement (Supplementary Fig. 4) showing the anti-Mcol4 antibody only detects immature cells (as previously described by Zenkert et al., 2011). Supplementary Fig 4 is referenced in the text as follows. The text of the caption is also provided below.

Lines 114-115: "By contrast, minicollagen-4 (Mcol4), which labels all types of stinging cells before maturation of the capsule (**Supplementary Fig. 4**), was unaffected."

Caption Supplemental Fig 4: "Detection of immature stinging cells with α -Mcol4 antibody in NvSox2 mutant animals. (A) α -Mcol4 labeled mutant stinging cells in the body wall of a tentacle bud stage animal shown in a 3D rendered z-stack (max projection) and an individual optical section (z-plane). Cells labeled with α -Mcol4 are immature and do not yet have a capsule that is visible in DIC (white circle) as previously shown for wild type *N. vectensis* ^{3,4}. Ecto – ectoderm, M – mesoglea. (B) α -Mcol4 labels immature stinging cells in the tentacle tip of a tentacle bud stage animal. A developing large piercing cell (basitrichous isorhiza nematocyte) is labeled with α -Mcol4 (white arrowhead); at this stage the capsule is

not yet visible with DIC (white circle). A mature mutant cell (orange arrowheads) has a capsule that can clearly be seen in DIC (false colored orange) but it not labeled with α -Mcol4 antibody.”

REVIEWERS' COMMENTS

Reviewer #1 (Remarks to the Author):

The authors have carefully revised their manuscript following the reviewers' suggestions. In particular, the identity of the cellular phenotypes observed after NvSox2 knock-down clearly demonstrate that the mutated phenotype corresponds to spirocysts. But also all other suggestions and comments were addressed. In sum, it is an excellent paper.

The homeotic conversion of one capsule type to another by the loss of a single transcription factor convincingly demonstrates how evolutionary novelties can arise by only one mutation. I expect that this work will receive much attention in the future and will be cited along with the famous homeotic transformations known from *Drosophila*.

Some (minor) points should be addressed and considered.

Title. The title indicates the phylogenetic relevance of the paper ("Single cell atavism reveals ..."). This is also true, as the authors show that spirocysts are the simpler capsule type compared to the more complex nematocysts with their barbed tubule. But spirocysts do not occur in Octocorallia, the basal group within Anthozoa, nor in the derived Medusozoa (jelly fish and hydroids). Accordingly, spirocysts should have been present at the base of Cnidaria but then got lost in several groups. I would also suggest that "cnidae" be included in the title, e.g., "Single-cell atavism of cnidocytes reveals.... "

References. In the introduction the cnidocyst / nematocyst is characterized as an "explosive organelle", but the original work on hydroid capsules unravelling these properties should be also cited (doi: 10.1016/j.cub.2006.03.089, doi: 10.1126/science.6695186., doi: 10.1016/j.tig.2008.07.001).

Reviewer #3 (Remarks to the Author):

The authors have addressed all comments raised in the first round of review. They have provided new diagrams that clarify things beautifully. They have also provided measurements that I think really drive home their message.

Comments from reviewer 1

1. "Title. The title indicates the phylogenetic relevance of the paper ("Single cell atavism reveals ..."). This is also true, as the authors show that spirocysts are the simpler capsule type compared to the more complex nematocysts with their barbed tubule. But spirocysts do not occur in Octocorallia, the basal group within Anthozoa, nor in the derived Medusozoa (jelly fish and hydroids). Accordingly, spirocysts should have been present at the base of Cnidaria but then got lost in several groups."

Response – The data presented in this manuscript do not really support this idea. In *Nematostella*, we show that only one type of piercing cell (small basitrichous isorhiza) appears to have evolved from an ancestral robust spirocyte; it is therefore unlikely that the robust spirocyte is the ground state of all other stinging cells across cnidarians. Rather, what we have shown is a mechanism by which any type of stinging cell could be co-opted to give rise to a stinging cell with a novel morphology. Figure 6d shows one very conservative interpretation of how the piercing cells in the body wall of Hexacorals (sea anemones, corals, and tube anemones) could have diversified but this is only a small part of the story. We greatly appreciate the depth with which Reviewer 1 has considered our ideas but without additional data we see no reason to suggest that all types of stinging cells evolved from a robust spirocyte.

2. "I would also suggest that "cnidae" be included in the title, e.g., "Single-cell atavism of cnidocytes reveals.... " "

Response - We deliberately left the cell type (cnidocytes) out of the title to make this story as appealing to a broad readership as possible. We think we may have discovered a fundamental (and broadly generalizable) mechanism for generating cell diversity (single-cell atavism) that applies to cell types outside of cnidocytes. We prefer to keep the title as written but will gladly defer to the editors on this style issue.

3. "References. In the introduction the cnidocyst / nematocyst is characterized as an "explosive organelle", but the original work on hydroid capsules unravelling these properties should be also cited (doi: 10.1016/j.cub.2006.03.089, doi: 10.1126/science.6695186., doi: 10.1016/j.tig.2008.07.001.

Response – We are grateful to Reviewer 1 for catching these omissions and have added these citations as suggested.